# Assessment of ICESat-2 ice surface elevations over the CHINARE route, East Antarctica, based on coordinated multi-sensor observations

Rongxing Li[1,2], Hongwei Li[1,2], Tong Hao[1,2*], Gang Qiao[1,2*], Haotian Cui[1,2], Youquan He[1,2], Gang Hai[1,2], Huan Xie[1,2], Yuan Cheng[1,2], Bofeng Li[1,2]

[1]Center for Spatial Information Science and Sustainable Development and Applications, Tongji University,
[2]College of Surveying and Geo-informatics, Tongji University, Shanghai 200092, China

*Correspondence to*: Tong Hao (tonghao@tongji.edu.cn), Gang Qiao (qiaogang@tongji.edu.cn)

**Abstract.** We present the results of an assessment of ice surface elevation measurements from NASA's Ice Cloud and land Elevation Satellite-2 (ICESat-2) along the CHINARE (CHINese Antarctic Research Expedition) route near the Amery Ice Shelf in East Antarctica. The validation campaign was designed and implemented in cooperation with the 36th CHINARE Antarctic expedition from December 2019 to February 2020. The assessment of the ICESat-2 geolocated photon product (ATL03) and land ice elevation product (ATL06) was performed based on coordinated multi-sensor observations using two roof-mounted kinematic GNSS receivers, two line arrays of corner cube retroreflectors (CCRs), two sets of retroreflective target sheets (RTSs), and two unmanned aerial vehicles (UAVs) with cameras. This systematic validation of the ICESat-2 data covered a variety of Antarctic ice surface conditions along the 520 km traverse from the coastal Zhongshan Station to the inland Taishan Station. This comprehensive investigation is complementary to the 750 km traverse validation of flat inland Antarctica containing a 300 km latitude traverse of 88 °S by the mission team (Brunt et al., 2021). Overall, the validation results show that the elevation of the ATL06 ice surface points is accurate to 1.5 cm with a precision of 9.1 cm along the 520 km CHINARE route. The elevation of the ATL03 photons has an offset of 2.1 cm from a GNSS-surveyed CCR, and is accurate to 2.5 cm with a precision of 2.7 cm as estimated by using RTSs. The validation results demonstrate that the estimated ICESat-2 elevations are accurate to 1.5–2.5 cm in this East Antarctic region, which shows the potential of the data products for eliminating mission biases by overcoming the uncertainties in the estimation of mass balance in East Antarctica. It should be emphasized that the results based on the CCR and RTS techniques can be improved by further aggregation of observation opportunities for a more robust assessment. The developed validation methodology and sensor system can be applied for continuous assessment of ICESat-2 data, especially for calibration against potential degradation of the elevation measurements during the later operation period.

# 1 Introduction

The new photon-counting laser altimetry satellite, Ice, Cloud, and Land Elevation Satellite-2 (ICESat-2), was successfully launched by the National Aeronautics and Space Administration (NASA) on September 15, 2018 (National Research Council, 2007; Markus et al., 2017; Neumann et al., 2019). It is a follow-up to the previous ICESat laser altimetry mission, which is based on the full waveform ranging (Zwally et al., 2002; Schutz et al., 2008). The primary scientific objective of the ICESat-2 mission is to determine ice sheet height changes through continuous measurements at scales from the outlet glaciers to the entire ice sheet (Markus et al., 2017). The primary instrument onboard ICESat-2, the Advanced Topographic Laser Altimeter System (ATLAS), is a photon-counting laser altimeter using 532 nm wavelength laser pulses, which is designed to conduct surface-elevation observations at centimeter-level accuracy (Markus et al., 2017; Neumann et al., 2018, 2019). We use Release 003 of the ICESat-2 geolocated photon elevation (ATL03) and land ice surface elevation (ATL06) products provided by the US National Snow and Ice Data Center (NSIDC) (NSIDC, 2021; Neumann et al., 2019; Smith et al., 2019).

The calibration and validation of measurements are important for all satellite missions, particularly for missions with new instruments or technology, such as the photon-counting laser altimeter on-board ICESat-2. A comprehensive validation of the surface elevations of the previous ICESat mission using ground Global Navigation Satellite System (GNSS) observations was performed on the Salar de Uyuni in Bolivia, which also estimated the inter-campaign biases occurred between different campaigns during the mission (Borsa et al., 2019). Prior to ICESat-2 launch, calibration and validation experiments were conducted on both the Greenland and Antarctic ice sheets (Brunt et al., 2017 and 2019b; Magruder and Brunt, 2018). The annual Antarctic campaigns traversed a 300 km stretch of the interior of Antarctica near 88 °S and intersected 20% of the ICESat-2 reference ground tracks (RGTs) (Brunt et al., 2019b). The results showed that surface-elevation biases for the tested altimeters, including the Multiple Altimeter Beam Experimental lidar (MABEL), over the flat ice sheet interior are less than 0.12 m with a precision of 0.09 m or better (Brunt et al., 2017). After launch, the mission team conducted a 750 km 88 °S traverse validation from December 31, 2018 to January 11, 2019 using kinematic GNSS to assess the elevation and horizontal accuracy of the ICESat-2 data products (Release 001) on the Antarctic Ice Sheet (AIS) (Brunt et al., 2019b). These results indicated that ATL03 data (Release 001) are accurate to 5 cm of surface elevation with a precision of 13 cm, while ATL06 data (Release 001) are accurate to 3 cm with a precision of 9 cm. The NASA-led team also used corner cube retroreflectors (CCRs) to collect ICESat-2 signatures at known points and determined the horizontal geolocation accuracy of the laser pointing as 2 - 5 m (Magruder et al., 2020), specifically ranging from 2.5 m for laser spot 6 to 4.4 m for laser spot 2 (Luthcke et al., 2021). Additionally, the average beam diameter was estimated as ~11 m (Magruder et al., 2020). Although this ICESat-2 validation campaign covered a long traverse of the flat Antarctic interior along the latitude of 88 °S, additional coverage containing the lower-latitude interior and coastal regions in AIS should make the validation of ICESat-2 data complete with an ample and comprehensive understanding of elevation of diverse regions of AIS. More specifically, such a validation in East Antarctica should help confirm the regional ICESat-2 surface elevation accuracy with which it is expected to reduce the

uncertainty of the mass balance and change rate in East Antarctica (Zwally et al., 2015; Scambos and Shuman, 2016; Richter et al., 2016).

To assess the accuracy of the Antarctic surface elevations provided in the ICESat-2 ATL03 and ATL06 data products and their capability for the estimation of volume changes in AIS, complementary to the current validation efforts by the mission team, we designed and implemented an independent validation campaign based on a set of coordinated multi-sensor experiments

along the 520 km 36[th] CHINARE (CHINese Antarctic Research Expedition) route in East Antarctica from December 2019 to February 2020 (Fig. 1). The ground sensors used in this coordinated validation campaign include two roving GNSS receivers on a snowcat, five GNSS master receivers on the ice surface, two line arrays of CCRs, two sets of retroreflective target sheets (RTSs), and two unmanned aerial vehicles (UAVs).

In order to validate the ATL03 and ATL06 data along the CHINARE route from the coastal Zhongshan Station to the inland

Taishan Station, two roving GNSS receivers of CHC i70 from CHC Navigation Technology LTD (https://www.chcnav.com/product-detail/i70-gnss, last accessed on April 12, 2021) were installed on roof of a snowcat, Pisten Bully Polar 300, to measure ice surface elevations using the post processed kinematic (PPK) positioning technique. Supported by the precise point positioning (PPP) technique, five GNSS base stations with CHC i70 receivers were deployed every ~100 km along the traverse to enable the PPK positioning of the vehicle. Two line arrays of ten upward-looking CCRs (optical

prisms) with known elevations were deployed at sites near Zhongshan Station and Taishan Station, respectively, to reflect photons for the verification of individual photons. We used one rectangular (5 m $\times$150 m) RTS for each site to investigate the reflectivity and elevation accuracy of photons reflected from selected RTS coatings. Finally, two UAVs, DJI Phantom 4 (https://www.dji.com/hk-en/phantom-4-rtk, last accessed on April 12, 2021), were used to acquire images for the generation of digital elevation models (DEMs) with centimeter accuracy (vertical) for an areal assessment of ICESat-2 surface elevations.

The real-time kinematic (RTK) positioning technique was applied to provide horizontal and vertical positions of the CCRs, GNSS points on RTSs, and control points for UAV - DEM reconstruction. The campaign initially included the Great Wall Station as the third site, where the CCRs and RTSs were planned to be used. Due to logistic difficulties during the Covid-19 pandemic, the experiments at this site were cancelled, which unfortunately affects the completeness of our planned geographic coverage. Nevertheless, our calibration and validation work at Zhongshan Station and Taishan Station can provide in-depth

knowledge of ICESat-2 surface elevation accuracy taking into account various terrain characteristics of coastal and inland Antarctica.

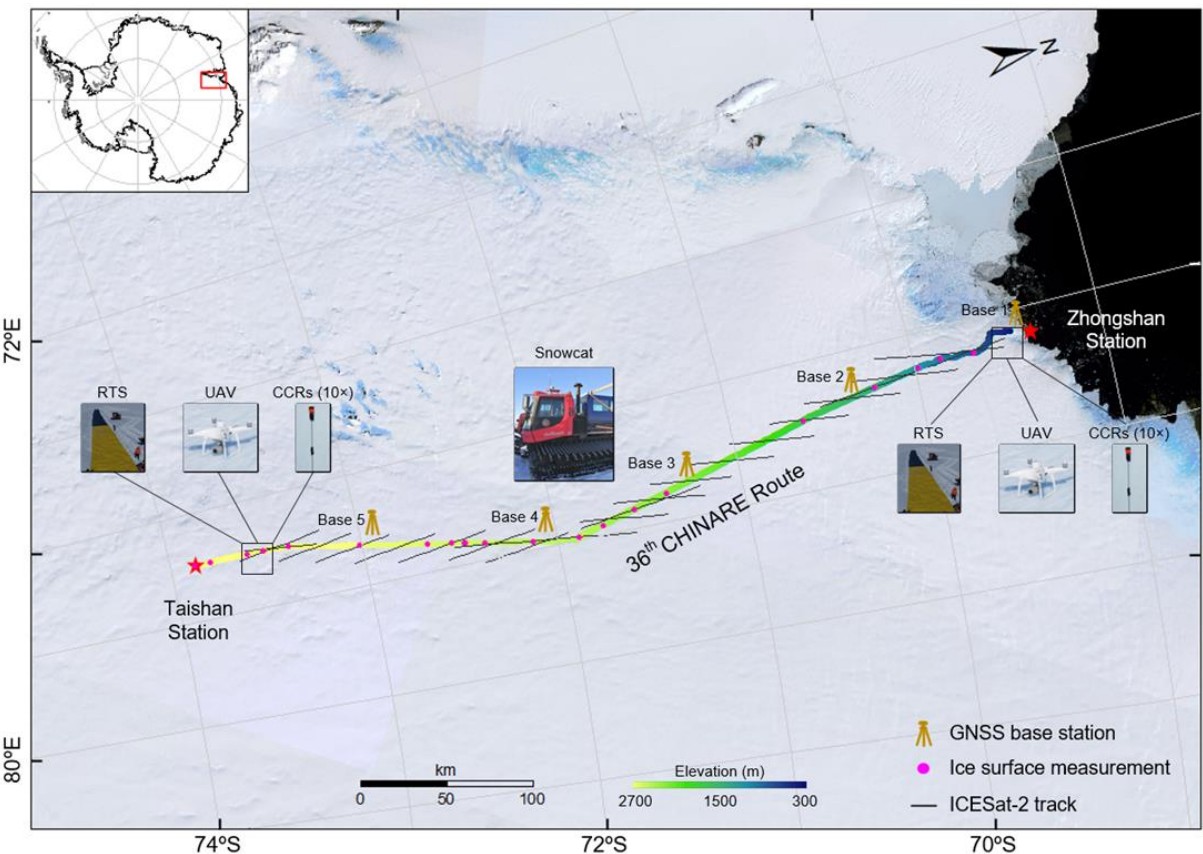

**Figure 1. ICESat-2 validation campaign based on coordinated multi-sensor ground observations along the 36th CHINARE route. The Landsat image mosaic of Antarctica (Bindschadler et al., 2008) is used as background.**

## 2 Data

### 2.1 ICESat-2 data

The ICESat-2 Release 003 data used for ground validation in this study are L2A Global Geolocated Photon Data (ATL03) and L3A Land Ice Height data (ATL06; Neumann et al., 2019; Smith et al., 2019) collected along 60 ascending and 78 descending tracks between November 10, 2019 and February 21, 2020 (Table 1).

**Table 1. Summary of the ICESat-2, GNSS and UAV observations used in this study.**

| Data type | Acquisition date (UTC) | Number of observations | Resolution | Geolocation accuracy | Format | Source |
|---|---|---|---|---|---|---|
| *Data for ICESat-2 ATL03 and ATL06 validation along GNSS traverse* | | | | | | |
| ICESat-2 ATL03 Release 003 | 2019-11-10 to 2020-02-21 | 20,562 photons for estimation | ~0.7 m spacing | planimetric: 2–5 m | HDF5 | NSIDC |
| ICESat-2 ATL06 Release 003 | 2019-11-10 to 2020-02-21 | 758 points for estimation | ~20 m spacing | not applicable | HDF5 | NSIDC |
| GNSS observations along traverse | 2019-12-10 to 2020-02-14 | 625,358 points | ~4 m spacing | vertical: 0.3 ± 5.8 cm | binary | this study |
| *Additional data for ICESat-2 ATL03 validation based on CCRs and RTSs* | | | | | | |
| GNSS observations at CCRs and RTSs near Zhongshan Station | 2020-01-22 | 137 points | at CCRs and distributed on RTSs | vertical: 1.3 cm | binary | this study |
| GNSS observations at CCRs near Taishan Station | 2020-01-23 | 10 points | at CCRs | vertical: ~1 m | binary | this study |
| *Additional data for ICESat-2 ATL06 validation based on UAV-DEM* | | | | | | |
| UAV image orthophoto near Zhongshan Station | 2020-01-22 | 154 scenes | ~4.6 cm | planimetric: 2.1 cm | GeoTIFF | this study |

## 2.2 GNSS data

We collected a total of ~1008 km (~516 km inbound and ~492 km outbound) of kinematic GNSS survey data during the 36[th] CHINARE inland expedition between Zhongshan Station and Taishan Station (Fig. 1), using two roving GNSS receivers mounted on roof of the snowcat (Fig. 4a). Antenna 1 served as the main GNSS receiver, while Antenna 2 also provided location data for an ice penetrating radar (IPR) mounted on the snowcat during the inbound trip. Due to incidental battery problems and inter-equipment interferences between Antenna 2 and IPR, the GNSS rover surveying was carried out by a combination

of two receivers. The GNSS receivers obtained data from the Global Positioning System (GPS), GLObal NAvigation Satellite System (GLONASS), Galileo Global Navigation Satellite System (Galileo), and BeiDou Navigation Satellite System (BDS). Along the inbound traverse journey (December 10–15, 2019), each GNSS base station was deployed on the ice surface before the snowcat started its ~100 km survey. Bases 2 – 5 with an interval of ~100 km collected data during the batteries' lifetime (~3 days). Base 1 was AC powered and located on the roof of a container near the runway of the Russian Progress Station.

During the outbound traverse (February 8–14, 2020), we only used Base 1 since the other four stations had run out of battery after the inbound traverse trip. The sampling rate was set to 1 Hz to conserve the power and storage space, and the elevation angle mask was set to 7 °to reduce the multipath effect. The same settings were also applied for the GNSS survey of the roving snowcat along the traverse, CCRs, RTS sheets and UAV – DEM near Zhongshan Station. Ten CCRs, 137 randomly distributed GNSS points on the RTS, and three ground control points (GCPs) for UAV geometric control near Zhongshan Station were

surveyed using the RTK technique. The observation time at each point was about 4 - 5 seconds. Due to logistic difficulties, CCRs near Taishan Station were surveyed using the single-point positioning technique at ~1 m accuracy level.

**2.3 CCR data**

Under the time constraints of equipment shipment before expedition and manufacturing cycle of new products we used ten readily available CCRs of 6 cm diameter (Fig. A1a) at each site, which were originally designed for ground - based laser

distance measurement. They were placed linearly at a 10 m interval across a nominal ICESat-2 ground track to reflect photons from ATLAS (Fig. 2a). To mitigate the error induced by the possible subsidence of the aluminum pole supporting each CCR, a wood base (40 cm $\times$ 40 cm $\times$ 1 cm) was assembled at the bottom of each pole. The elevations of the nominal centers of the CCR lens were designed to vary within one meter for identification of individual CCRs from which the signal was reflected. The poles were manufactured before the expedition with different lengths. The actual elevations of CCRs were influenced by

the ice surface topography where they were deployed (Table C1).

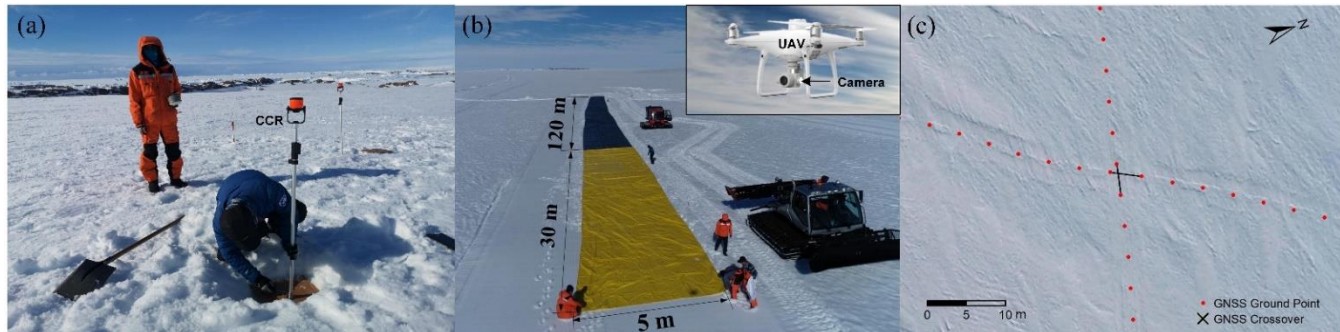

**Figure 2. (a) Ten CCRs were installed along a line across an ICESat-2 ground track at each site; (b) a two-coating RTS (5 m $\times$ 150 m with yellow and dark green coatings) was deployed near Zhongshan Station and surveyed by a UAV; and (c) example of a GNSS crossover point and snowcat tracks on a UAV image.**

**2.4 RTS data**

RTSs were designed to assess the reflectivity and elevation accuracy of the ICESat-2 photons from a known artificial surface in the Antarctic environment. We deployed an RTS of 5 m $\times$ 150 m at each site, which was oriented perpendicular to the two ICESat-2 tracks (weak and strong) that are separated by 90 meters. The coatings of the RTSs were selected from among 28 candidates through a pre-expedition experiment. For each coating, an Avafield-3 spectrometer

(https://www.avantes.com/products/spectrometers/compactline, last accessed on April 12, 2021) was used to measure the reflectivity ($R$) of each coated RTS at the ICESat-2 wavelength of 532 nm. We also used a lidar ranger of Riegl BDF-1 (http://www.riegl.com/uploads/tx_pxpriegldownloads/RIEGL_BDF-1_Datasheet_2019-05-31.pdf, last accessed on April 12, 2021) with a laser wavelength of 532 nm to measure the corresponding reflectance ($r$). We selected a silver-gray coating with $R = 0.235$ as it was the closest to the reportedly highest estimated probability (EP) of photon detection coating with $R = 0.28$

(Hartzell et al., 2018). We added two other coatings, i.e., yellow ($R = 0.532$) for high reflectivity and dark green ($R = 0.060$) for low reflectivity. The RTS near Zhongshan Station had yellow and silver–gray coatings, while the one near Taishan Station had yellow and dark green coatings.

**2.4 UAV data**

A UAV equipped with an HD camera (1″ CMOS, 20 megapixel) was flown over an area of 750 m $\times$ 500 m near Zhongshan
Station. A total of 154 images were collected with an along-track overlapping of 80% and a side overlapping of 75%. At a
flight height of ~250 m the ground resolution was 4.6 cm. The CCRs and RTS were inside the mapping area.

**3 Method**

As shown in Fig. 3, this Antarctic validation campaign was designed to achieve three goals: a) assessment of ICESat-2 surface
elevations (ATL03 and 06) using a kinematic GNSS survey along the 520 km CHINARE route (69.46–73.86ºS); b) assessment
of elevations of the targeted ICESat-2 photons (ATL03) from the CCRs and RTSs; and c) assessment of ICESat-2 elevations
(ATL06) using a UAV survey. The major steps of processing are illustrated in Fig. 3.

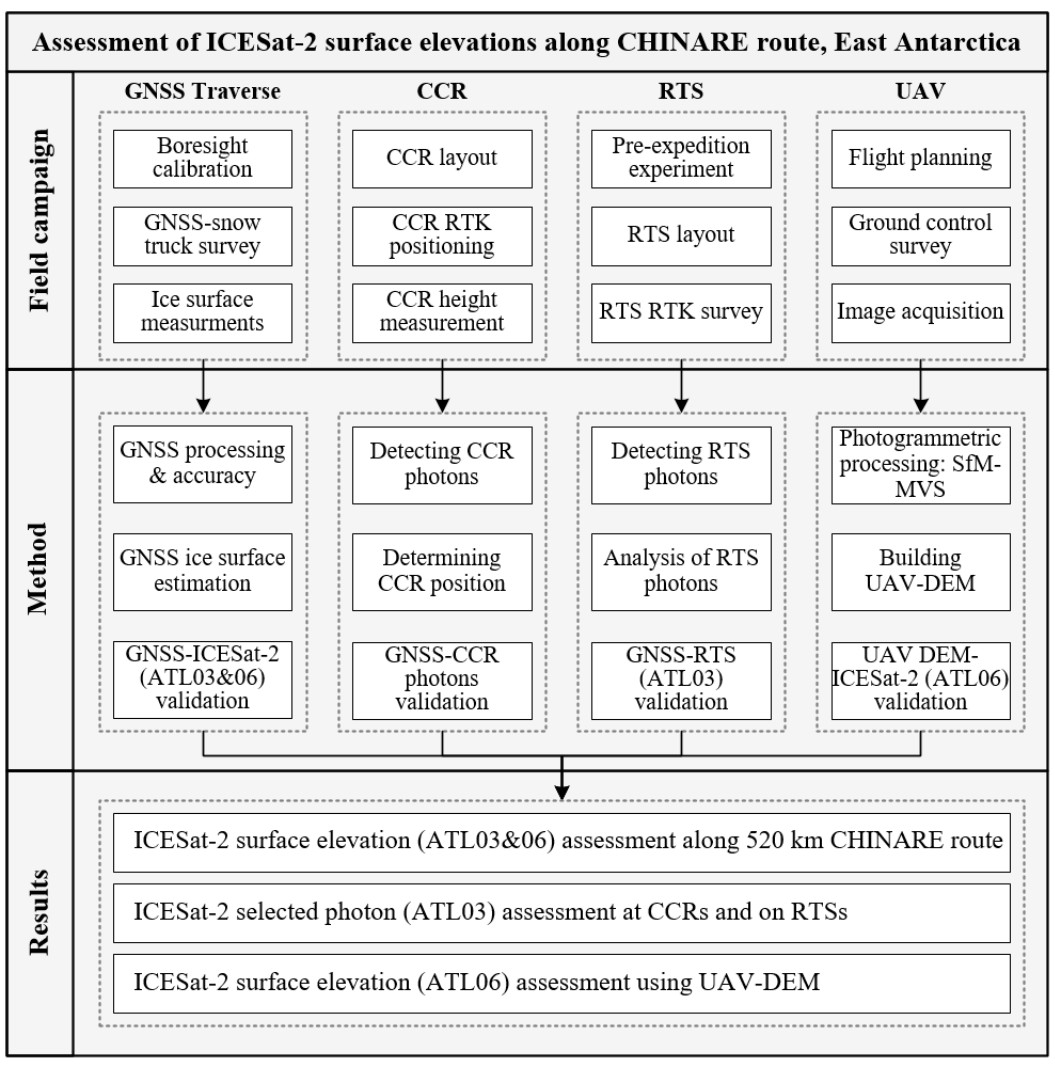

**Figure 3. Flow chart for assessment of ICESat-2 ice sheet surface elevations along the CHINARE route as well as two validation sites, East Antarctica, based on coordinated multi-sensor observations.**

### 3.1 GNSS data processing

The double-frequency data of the base stations were post processed using the PPP technique implemented in the software system of multi-frequency and multi-system instantaneous PPP (MUSIP) developed at Tongji University (Li et al., 2019), with the precise ephemeris and clock products provided by GFZ (ftp://ftp.gfz-potsdam.de/GNSS/products/mgex). Two roving receivers, Antenna 1 and Antenna 2, were mounted on the roof of the snowcat (Fig. 4a). Combined with the PPP results of the base stations, GNSS data collected by the roving receivers along the traverse were processed using the PPK positioning technique that is implemented in an open source software package RTKLIB version 2.4.3 (http://www.rtklib.com/, last accessed on April 12, 2021) developed by the Laboratory of Satellite Navigation at Tokyo University of Marine Science and Technology (Takasu et al., 2009; Takasu, 2013). The snowcat positions within a segment of up to ~100 km from each base station were estimated.

The internal precisions of the estimated positions from the PPP and PPK processing are given by the software systems. Furthermore, we used the accuracy computed from elevation differences at crossovers where the GNSS surveyed track intersected itself, as shown in Fig. 2c. These crossovers are the intersections of tracks by the snowcat which occurred usually during instrument installations, observations, and overnight breaks. Within a neighborhood of the intersection we fit two lines to compute the crossover location and elevation difference (Kohler et al., 2013).

The RTK positioning technique is applied to estimate positions of the CCRs and GNSS points on the RTS sheets near Zhongshan Station. We used a known GNSS control point at Zhongshan Station as a reference point for RTK. The CCR and RTS checkpoint positions were estimated in real-time by the GNSS receiver's onboard software (https://www.chcnav.com/uploads/i70_DS_EN.pdf, last accessed on April 12, 2021). Additionally, the UAV - DEM reconstruction was geometrically controlled using three GNSS GCPs. The positions were estimated by the RTK technique implemented in the UAV package (https://www.dji.com/hk-en/phantom-4-rtk, last accessed on April 12, 2021). The accuracy of the RTK positions is estimated based on internal precisions given by the applied GNSS systems and the accuracy of the reference point.

In the ICESat-2 products geographic coordinates (latitude and longitude) are defined based on the WGS84 ellipsoid and heights are referenced to the ITRF2014 frame (Brunt et al., 2019; Neumann et al., 2019); corrections for solid earth tides, ocean loading, solid earth pole tide, ocean pole tide and others are applied to the ATL03 data (Neumann et al., 2019). On the other hand, the processed GNSS data are also referenced based on the WGS84 ellipsoid (Schröder et al., 2017); the ITRF2014 reference frame is used in the GFZ precise ephemeris and precise orbit products which is input to the RTKLIB and MUSIP post processing systems; furthermore, the geophysical corrections for the above tides are applied (Petit and Luzum, 2010). Thus, the reduced ice surface elevations are given as "tide-free" (Neumann et al., 2019) and the permanent crustal deformation is removed (Schröder et al., 2017; Brunt et al., 2021).

## 3.2 Kinematic GNSS – ICESat-2 ice surface elevation validation

### 3.2.1 Derivation of ice surface elevations from GNSS observations

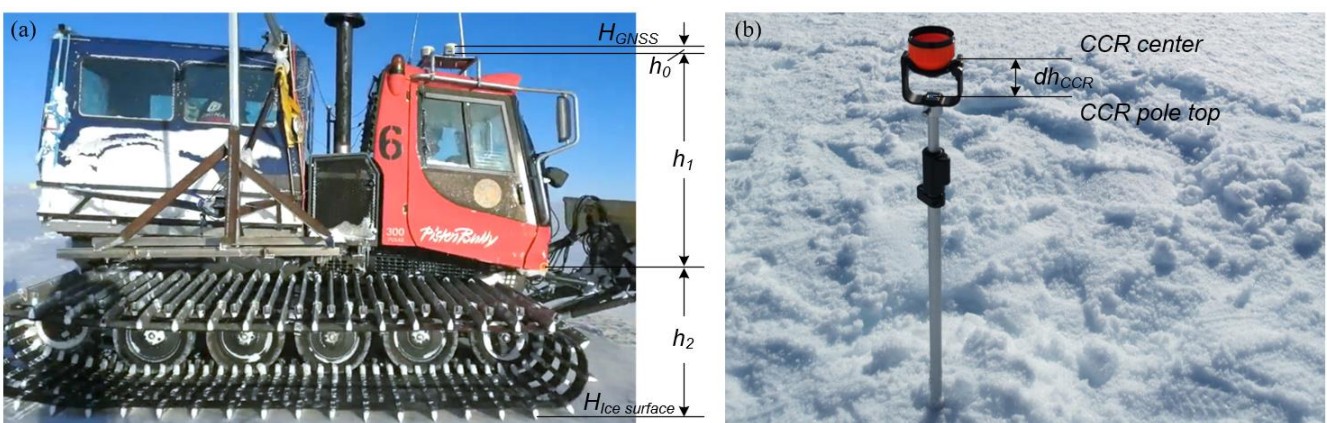

Figure 4. (a) GNSS roving receivers mounted on the roof of a Pisten Bully snowcat and boresight parameters for ice surface elevation estimation from the GNSS observations, and (b) CCR installed in the field with the pole top elevation ($h_{Pole\ top}$) surveyed by GNSS and the CCR center height ($dh_{CCR}$) measured using a steel tape.

Given the antenna phase-center elevation of a roof-mounted GNSS roving receiver $H_{GNSS}$ (Fig. 4a), the elevation of the ice surface can be computed as

$$H_{Ice\ surface} = H_{GNSS} - h_0 - h_1 - h_2 , \tag{1}$$

where $h_0$ is the antenna phase-center height above the mounting plane and $h_1$ is the vertical distance between the mounting plane and a reference point marked by a yellow dot on the snowcat in Fig. 4a, from which the vertical distance to the ice surface $h_2$ can be measured. $h_0$ was given by the manufacturer as 10.1 cm. Prior to the GNSS traverse survey, a boresight calibration was performed to estimate the fixed parameter $h_1$. We used a SOKKIA CX-102LN total station to measure $h_1$ three times and estimated an average of 191.1 cm. During the breaks along the traverse we used a steel tape to measure $h_2$. To reduce measurement error due to the uneven ice surface, we measured it three times across a section with an interval of 10 cm. Thus, wherever $h_2$ is measured, the ice surface elevation $H_{Ice\ surface}$ can be computed from the roving receiver observation $H_{GNSS}$ and the boresight parameters $h_0$ and $h_1$. At other traverse points between these direct ice surface measurements, we interpolate the values of $h_2$ using inverse distance weighting (IDW).

### 3.2.2 Ice surface elevation comparison along the GNSS traverse

Each ICESat-2 orbit has three pairs of ground tracks (1, 2 and 3 in Fig. 5a) which are 3.3 km apart (ICESat-2 Technical Specs, 2020). Each track pair corresponds to two laser beams, i.e., weak beam and strong beam. The left and right correspondence in Fig. 5a may change as the spacecraft changes its orientation. Furthermore, the reference ground track (RGT) is defined as an imaginary track between the nadir ground track pair (2L and 2R). All six laser beams then have laser spot IDs (1, 2…, 6) which

are independent of spacecraft orientation. During our study period, the correspondences are (Laser spot 1: 3R), (Laser spot 2: 3L), (Laser spot 3: 2R), (Laser spot 4: 2L), (Laser spot 5: 1R), and (Laser spot 6: 1L) (Neumann et al., 2019). For example, an ascending ICESat-2 orbit (six tracks) intersected the CHINARE route (red line) at ~420 km from Zhongshan Station (Fig. 5b). In the enlarged area of the intersection (Fig. 5d), we select the ICESat-2 ice surface points (ATL06) and photons (ATL03) along track 2R (~80 m for ATL06 and ~40 m for ATL03). Specifically, to reduce the impact of non-signal and noisy

measurements we select the ATL06 land ice elevation measurements using the *atl06_quality_summary* flag (*best_quality*) and the ATL03 geolocated photons using the *signal_conf_ph* flag (*medium and high*). We also consider the ATL03 photons with "*signal_conf_ph*" equal to *buffer and low* to reduce the effect of the transmit pulse shape bias that may be caused by truncation of the return pulse through exclusion of these lower confidence photons (Smith et al., 2019; Brunt et al., 2019b). Furthermore, these should be collected within 30 days of the GNSS campaign period to avoid significant changes of the ice surface and

other environmental conditions between the two data sets. In order to estimate the ice surface elevations ($H_{Ice\ surface}$) at the GNSS traverse points using Equation (1), the height from the snowcat to the ice surface ($h_2$) is measured at the locations when the snowcat stops during campaign breaks.

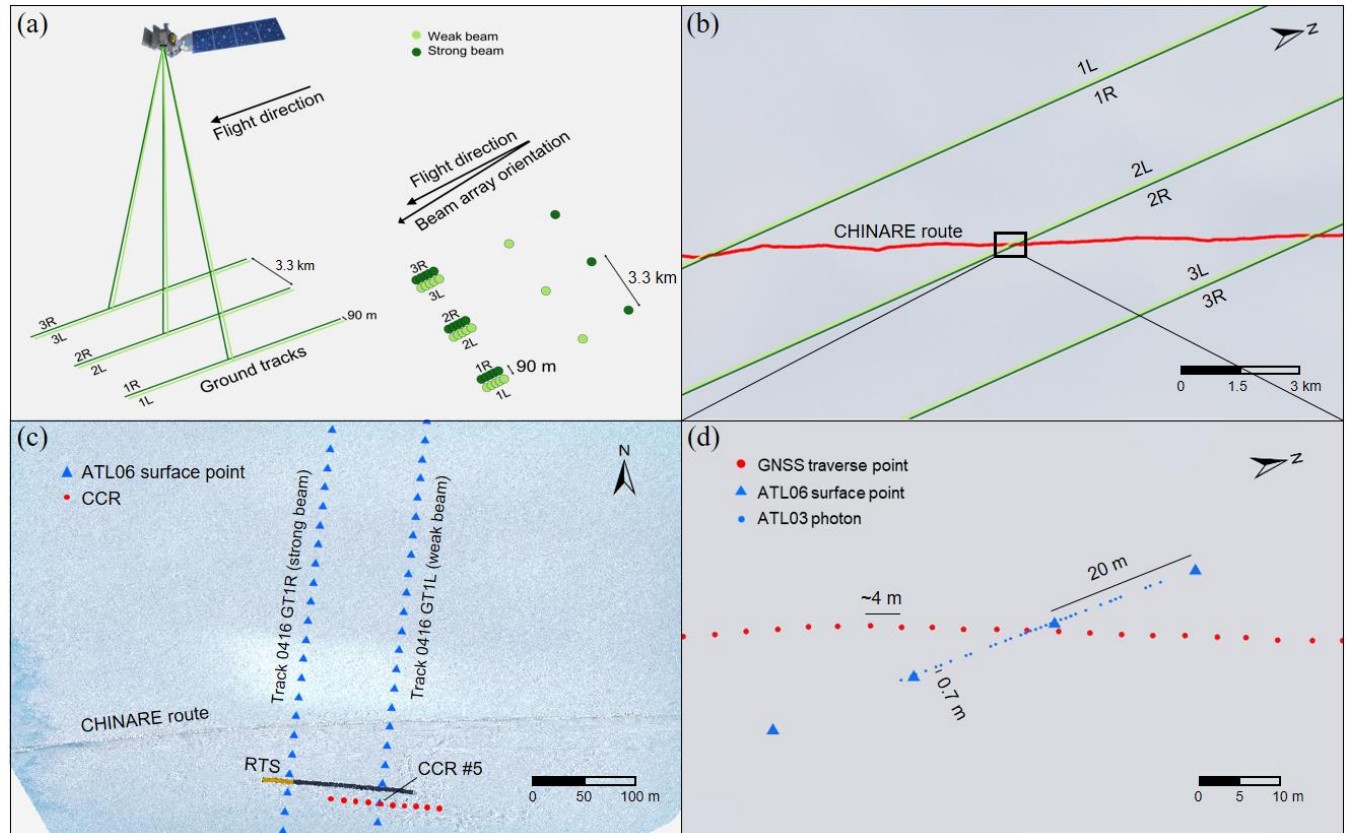

**Figure 5. (a) ICESat-2 ground tracks and beam pattern, modified from ICESat-2 Technical Specs (2020), (b) Ground tracks intersecting the CHINARE route (red line) at ~420 km from Zhongshan Station, (c) UAV - DEM mapping area and two ground**

**tracks near Zhongshan Station with a UAV orthophoto as background, and (d) ATL06 ice surface points and ATL03 photons (track 2R) and GNSS traverse points in the enlarged area of the rectangle in (b).**

Within an intersection area (Fig. 5d), the elevations of the GNSS traverse points are compared with the ICESat-2 ATL06 and
ATL03 data. First, at each ATL06 ice surface point we open a 20 m radius circle inside which there should be at least five GNSS points. We then calculate the difference between the ATL06 point elevation and the median ice surface elevation of the GNSS points. Secondly, for each ATL03 photon we find its nearest GNSS point within a 4 m wide search window to form a pair and calculate their elevation difference. Finally, if the number of pairs is sufficient ($\geq$ 30), we use the median among them to be the representative elevation difference between the ATL03 and GNSS data at this intersection.

**3.3 Validation of ICESat-2 photons using CCR and RTS elevations**

Design of a photon-capturing CCR array requires a number of considerations. The spacecraft orbiting and sensor pointing capability has been improved after the initial mission (Brunt et al., 2019a). Our preliminary analysis of Releases 001 and 002 data indicated that the offsets between the actual and reference ground tracks in the study area were reduced from up to ~3000 m during the initial mission, which is caused by a reference frame mismatch in the onboard software for the star cameras
(Luthcke et al., 2021), to 1–6 m before our expedition. Given a footprint of ~11 m diameter every ~0.7 m along the track (Markus et al., 2017; Magruder et al., 2020), ATL03 data have an average number of photons (confidence flag: *low-high*) of ~2 for weak beams and ~7 for strong beams per pulse on the ice surface in our study area; with a CCR field of view (FOV) of $\pm 35\,^{\circ}$ (see Appendix A), a single line CCR array across a track ensures along-track photon illumination. In the cross-track direction we installed 10 CCRs spaced every ~10 m across one weak (or strong) beam track (Fig. 5c); this ensures that at least
one CCR is placed inside a footprint of ~11 m along one track. Each CCR is designed to have a unique elevation for distinguishing the target CCR from other CCRs (Fig. 4b). The elevation of the CCR pole top ($h_{Pole\ top}$) is surveyed by using the precision RTK GNSS positioning method; then, the height from the CCR center, marked by the manufacturer as the theoretical photon bounce point, to the CCR pole top ($dh_{CCR}$) is measured using a steel tape. Thus, the elevation of the CCR is estimated as $h_{CCR} = h_{Pole\ top} + dh_{CCR}$. The field deployment and survey are carried out a few hours before the expected satellite pass to
avoid severe sinkage of the CCR.

The photons reflected from a CCR and received by ATLAS are represented as a streak of elevations along a track in the ICESat-2 ATL03 data (green dots in Fig. 6a); they are distributed on both sides of the GNSS-surveyed location (black square). We select the photons in the central section of the streak, approximately one footprint long (~11 m, within the red rectangle), as presented in inset of Fig. 6a to estimate the representative CCR photon position. In each pulse the photons are selected using
the confidence flag "*signal_conf_ph*" equal to *medium* (3) or *high* (4) and averaged to reduce noises and potential atmospheric effect. The average elevations (blue dots in inset) are used to fit a Gaussian curve. The peak of the curve is treated as the representative CCR photon position. An offset is then calculated between the CCR positions estimated from the ATL03 photons (peak point) and the GNSS survey (black square).

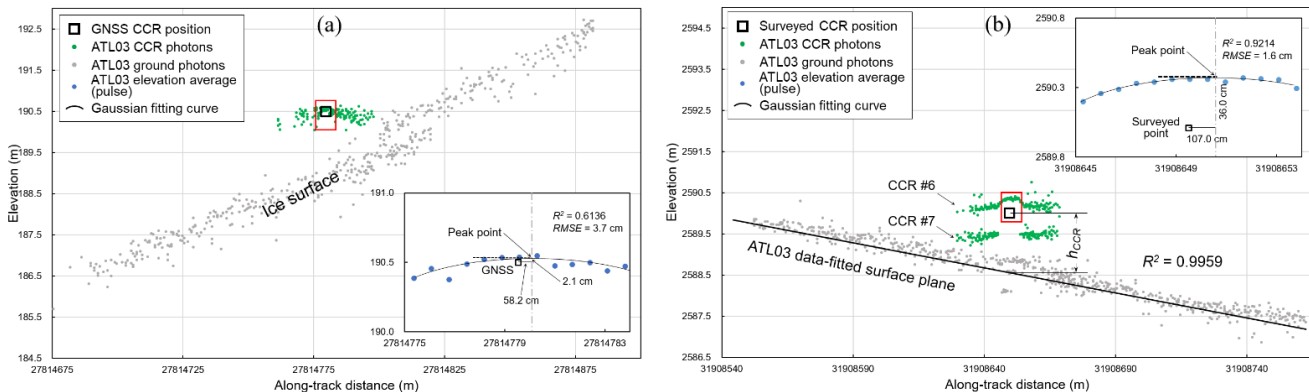


Figure 6. (a) CCR experiment near Zhongshan Station: returned CCR photons (ATL03, green dots), GNSS-surveyed CCR position (black square), and ice surface photons (ATL03, gray dots); inset: elevations averaged in each pulse (blue dots) in the enlarged red rectangle in (a) for fitting a Gaussian curve; and (b) CCR experiment near Taishan Station: returned CCR photons (ATL03, green dots), steel tape-surveyed CCR position (black square), and $h_{CCR}$ - height between CCR center and ice surface

(ATL03, gray dots); inset: elevations averaged in each pulse (blue dots) in the enlarged red rectangle in (b) for fitting a Gaussian curve.

Similarly, the high-quality photons reflected from an RTS sheet are selected from the ATL03 data to compare with the GNSS RTK points within a close neighborhood. Based on their elevation differences at all paired points the estimated bias and precision are given as the elevation validation result. Additionally, a radiometric experiment is performed to analyze the

influence of different RTS coatings on reflectance of the ICESat-2 photons. The total number of ATL03 photons is counted from all pulses along a track within the RTS sheet. These numbers are analyzed according to strong-weak beams, variant coatings, and firn on nearby ice surface. Furthermore, the full saturation fraction (FSF) flag "*full_sat_fract*", ranging from 0 to 1, represents the ratio of fully saturated pulses over a 20 m along-track segment (Neumann et al., 2020). The FSF values are considered to examine the relationship between the photon reflectance and effect of coatings.

### 3.4 Validation of ICESat-2 ice surface elevations using UAV - DEM

In the mapping area a set of ground control points (GCPs) are surveyed using the RTK positioning technique. They are further used to perform a photogrammetric absolute orientation (McGlone, 2013). The 3D surface points are reconstructed from the UAV images by using the structure-from-motion multi-view stereo (SfM-MVS) algorithm (James and Robson, 2012; Turner et al., 2014) implemented in the Pix4Dmapper software (version 4.5.6, https://support.pix4d.com/hc/en-

us/categories/360001503192-Pix4Dmapper, last accessed on April 12, 2021). As the result, a UAV - DEM and an orthophoto at a centimeter level accuracy (both horizontal and vertical) are generated. Thereafter, we evaluate the elevation differences $\Delta H$ between the elevations of the ICESat-2 ATL06 ice surface points ($H_{Ice\ surface}$) and the corresponding elevations of the UAV - DEM ($H_{UAV\_DEM}$):

$$\Delta H = H_{Ice\ surface} - H_{UAV\_DEM}. \tag{2}$$

## 4 Results

### 4.1 Kinematic GNSS – ICESat-2 ice surface elevation validation

The elevation precision values given by the GNSS PPP processing software system for the five base stations, from Base 1 to Base 5, are at the sub-centimeter (0.5 cm, 0.9 cm, 0.8 cm, 0.7 cm, and 0.9 cm) level. For the positions of the roof-mounted GNSS receivers along the inbound and outbound traverses, a threshold of the precision values given by the GNSS PPK positioning software system was used to filter out traverse points with large errors originating from rough terrain features and other noises, i.e., 3σ for the undulated topography near the coast (0–67 km) and 2σ for the relatively flat inland topography (67–520 km). As a result, a total of 625,358 GNSS traverse points were obtained, which have an average internal elevation precision of 1.6 ±0.6 cm given by the software system. Finally, the elevation accuracy of the GNSS traverse was assessed as 0.3 ±5.8 cm by using 26 crossovers of the traverse itself (Fig. 2c) where the GNSS surveyed elevations from two intersecting traverse segments were compared.

There were 20 locations along the traverse (Fig. 1) where $h_2$ was measured (Fig. 4a). Three outliers ($h_2$ measured to bottom of wheel print) at the beginning of the inbound traverse were eliminated. Along with other two bore sight parameters, $h_0$ and $h_1$, these direct measurements were used to derive the ice surface elevations $H_{Ice\ surface}$ from the roof-mounted kinematic GNSS observations $H_{GNSS}$. The average of the measured $h_2$ is 94.0 cm with a standard deviation of 2.8 cm. This variation is mainly attributed to the microtopography and firn density changes at different locations along the 520 km traverse from the coast to the highland interior. Out of the 134 intersections between the GNSS traverse and ICESat-2 tracks, we selected 60 intersections that are within 5 km of the $h_2$ direct measurement locations to enhance the comparability between the elevations observed by the ICESat-2 satellite and the kinematic GNSS receivers along the traverse. The average distance between the $h_2$ measurements and GNSS – ICESat-2 intersections is ~2366 m. We validated the elevations of the ICESat-2 ATL06 ice surface points and ATL03 photons using the GNSS-surveyed elevations that are summarized according to six ICESat-2 tracks separately (Table 2).

315

**Table 2. Assessment of ICESat-2 ATL06 ice surface points and ATL03 photons using the GNSS PPK technique with direct ice surface measurements (within 5 km of $h_2$) along the 36th CHINARE traverse. Bias and precision were estimated from their elevation differences using N ice surface points or photons. The difference is calculated as ICESat-2 elevation minus GNSS elevation.**

| Ground track (Laser spot ID) | ATL06 Bias ± Precision (cm) | ATL03 Bias ± Precision (cm) |
|---|---|---|
| GT1L (Laser spot 6) | +2.7 ±9.6 (N = 64) | +5.9 ±5.9 (N = 1518) |
| GT1R (Laser spot 5) | +3.0 ±7.3 (N = 62) | +1.7 ±6.7 (N = 2608) |
| GT2L (Laser spot 4) | +0.7 ±7.9 (N = 48) | -0.5 ±6.7 (N = 862) |
| GT2R (Laser spot 3) | - 2.3 ±12.0 (N = 42) | +5.8 ±14.0 (N = 1356) |
| GT3L (Laser spot 2) | +1.3 ±8.4 (N = 33) | +4.2 ±7.7 (N = 800) |
| GT3R (Laser spot 1) | - 0.7 ±8.7 (N = 36) | +4.6 ±10.9 (N = 2695) |
| ALL | +1.5 ±9.1 (N = 285) | +4.3 ±8.5 (N = 9839) |

Compared to the kinematic GNSS elevation observations, the ATL06 ice surface points have median elevation differences (bias) for the six ICESat-2 tracks ranging from -2.3 cm to 3.0 cm and precision values (1σ) ranging from 7.3 cm to 12.0 cm, resulting in an overall bias of 1.5 cm and precision of 9.1 cm. Similarly, the ATL03 photons have an overall bias of 4.3 cm and precision of 8.5 cm. No significant elevation differences were found between the tracks of the weak and strong beams. The difference between the bias of 1.5 cm for the processed ATL06 application product (L3A Land Ice Height data) and that of 4.3 cm for the unprocessed ATL03 product (L2A Global Geolocated Photon Data) is considered insignificant, taking their precision values, 9.1 cm and 8.5 cm, respectively into account.

We further extended our assessment to all intersections of the ICESat-2 tracks and the GNSS traverse without the above 5 km selection constraint. At each intersection, the $h_2$ value was calculated between two measurement locations using the IDW interpolation method. As shown in Table B1, the ATL06 and ATL03 data present a bias of 0.5 cm and 3.4 cm, respectively, which are comparable to these in Table 2. However, the overall precision values of 12.7 cm (ATL06) and 11.5 cm (ATL03) in Table B1 are relatively larger than 9.1 cm (ATL06) and 8.5 cm (ATL03) in Table 2.

## 4.2 Validation of ICESat-2 photon elevations using CCRs and RTSs

ICESat-2 passed across the line array of 10 CCRs near Zhongshan Station along RGT 0416 at 23:49 UTC on January 22, 2020 (Fig. 5c). About six hours before the ICESat-2 pass, the CCRs were deployed and RTK GNSS was surveyed. Based on the internal precisions of individual CCRs given by the GNSS system and accuracy of the known GNSS reference point, the elevation accuracy of ten CCRs is 1.3 cm. A GNSS sinkage survey was performed over 2 days and estimated a negligible sinkage of less than 1 cm. The ATL03 data showed a streak of 137 photon returns classified as medium to high quality in 50 pulses, spanning ~35 m along the weak beam track (1L; green dots in Fig. 6a). On the other hand, ICESat-2 passed across the CCR line array near Taishan Station along the weak beam track (2L) of RGT 0424 at 12:37 UTC on January 23, 2020 (green dots in Fig. 6b). Due to logistic difficulties, CCR positions were surveyed using the single-point positioning technique at ~1 m accuracy level in both the horizontal and vertical directions.

Among the CCRs, CCR #5 near Zhongshan Station and CCR #6 near Taishan Station were closest (~0.3 m and ~1.3 m) to the ground tracks and are called nadir CCRs. Given these CCR shifts from ground tracks, the uncertainty of the ground tracks themselves of up to ~6.5 m (Magruder et al., 2020), and the CCR interval of ~10 m, one to two CCRs may fall into a footprint of ~11 m. Fig. 6b clearly shows returned photons (green dots) of two layers with an elevation difference of ~50 cm. Hence, the photons from the nadir CCR (#6) with a higher signal level of the central disc of the Fraunhofer diffraction pattern (Smith et al., 2019; Magruder et al., 2020) were received by ATLAS within a window of ~9 m of the central section of the streak (upper layer, red rectangle in Fig. 6b), while those from the neighboring CCR (lower layer) with a lower signal level from the lobes were not present inside the window. Therefore, we only use the returned photons inside a window of ~9 m around the nadir CCRs to validate ICESat-2 photon elevations (insets of Figs. 6a and 6b).

From the returned photons near Zhongshan Station (Fig. 6a) we selected 51 photons of 13 pulses, 3.9 photons per pulse, located in the central part of the weak beam streak (Laser spot 6, ~9 m, inside the red rectangle). Both the track location and elevations of the photons matched those of CCR #5 (black square), suggesting that it was illuminated during the ICESat-2 pass. The elevations of the photons within each pulse, pulse id (/gtx/heights/ph_id_pulse) from 133 to 145, were averaged and used to fit a Gaussian function curve with an $R^2$ of 0.6136 and a fitting RMSE of 3.7 cm (inset of Fig. 6a), whose peak point was used as the position of the representative photon of the CCR. The offset between the peak point and the GNSS-surveyed position is 58.2 cm in the horizontal direction and 2.1 cm in the vertical direction.

CCR #6 near Taishan Station was found to have returned 52 photons in 13 pulses, 4 photons per pulse, from the weak beam track (Laser spot 4) within the ~9 m window in Fig. 6b. To estimate the CCR #6 elevation that is more accurate than the meter-level GNSS result, we first used the ATL03 ice surface photons (black dots in Fig. 6b) to fit a terrain surface plane with an $R^2$ of 0.9959. Using the measured CCR height $h_{CCR}$ in Fig. 6b and the fitted ice surface, the improved CCR elevation (black square) was calculated. After that, the photons were averaged within each of pulse (pulse id from 19 to 31, blue dots in inset of Fig. 6b) to fit another Gaussian function with an $R^2$ of 0.9214 and an RMSE of 1.6 cm. The peak position of the Gaussian function was used as the representative photon of the CCR that has an offset of 107.0 cm in the horizontal direction and 36.0 cm in the vertical direction from the estimated CCR location.

ICESat-2 passed across the RTS and the CCR line array along the same orbit at each site near Zhongshan Station and Taishan Station, respectively. There are seven check points on the yellow sheet near Zhongshan Station, which were surveyed using the GNSS RTK method with an accuracy of 1.3 cm. Pairing of the check points with the corresponding ATL03 photons was achieved with a maximum in-between distance of 1.22 m. The comparison between the two datasets resulted in an elevation uncertainty of $2.5 \pm 2.7$ cm.

The combinations of the different coatings and weak and strong beams are listed in the first column of Table 3. The three deployed RTS coatings were tested prior to the Antarctic expedition and their reflectivity was measured as 0.532 for yellow, 0.235 for silver–gray, and 0.060 for dark green coatings, respectively. The 5 m wide (7 pulses), high-reflectivity yellow coating reflected photons from the strong beam. There were in average 9.0 and 7.0 ATL03 photons (confidence flag: low-high) per pulse reflected within the 5 m RTS near Zhongshan Station and Taishan Station, respectively. Furthermore, as shown in Table

3, FSFs of two 20 m segments containing the yellow coating RTS sheets at the two sites are 0, indicating that the pulses were not saturated. For comparison, the average photon counts of 6.1 and 6.7 per pulse near the two sites were estimated over the

same sized bare ice surfaces (300 m away from RTS and CCR) along the same strong beam tracks. The FSF values were also 0. Therefore, the footprints of the high-reflectivity yellow coating RTSs showed similar photon reflectivity characteristics to the nearby bare ice surface. On the other hand, the two lower-reflectivity coatings, i.e., silver–gray and dark green coatings, reflected the lowest and highest numbers of average photon counts per pulse (1.1 and 9.7) from the weak beam, but showed higher FSF values (0.321 and 0.429). In contrast the ice surface nearby showed lower average numbers of reflected photons

per pulse (1.7 for both sites) and almost no saturated pulses with FSF of 0 and 0.072, respectively. We suggest that this abnormal phenomenon be attributed to contamination by the photons reflected from the illuminated CCRs, which were only ~9 m away from the RTSs (Fig. 5c).

**Table 3. Received photons and full saturation fractions (FSF) with respect to different RTS coatings.**


| RTS coating (ICESat-2 beam, site) | Pre-expedition reflectivity | Avg count per pulse (RTS) | Full saturation fraction (RTS) | Avg count per pulse (firn) | Full saturation fraction (firn) |
|---|---|---|---|---|---|
| **Yellow** (strong beam, Zhongshan) | 0.532 | 9.0 | 0 | 6.1 | 0 |
| **Yellow** (strong beam, Taishan) | 0.532 | 7.0 | 0 | 6.7 | 0 |
| **Silver–gray** (weak beam, Zhongshan) | 0.235 | 1.1 (CCR) | 0.321 (CCR) | 1.7 | 0 |
| **Dark green** (weak beam, Taishan) | 0.060 | 9.7 (CCR) | 0.429 (CCR) | 1.7 | 0.072 |

In summary, one CCR of the precisely GNSS-surveyed line CCR array near Zhongshan Station was illuminated. The estimated representative CCR photon has an offset of 2.1 cm in elevation and 58.2 cm in horizontal direction from the GNSS-surveyed CCR position. Seven precisely GNSS-surveyed check points that are evenly distributed on a highly reflective RTS near

Zhongshan Station were used to validate the elevations of their corresponding ATL03 photons (strong beam) and resulted in a bias and precision of 2.5 $\pm$ 2.7 cm.

### 4.3 Validation of ICESat-2 ice surface elevations using UAV - DEM

The 750 m $\times$ 500 m UAV - DEM mapping area near Zhongshan Station contains the RTS and CCR line array (Fig. 5c). ICESat-2 RGT 0416 passed through the area 8 h before the UAV image acquisition. Based on the internal precision given by the GNSS

system and accuracy of the known GNSS reference point, the elevation accuracy of three GCPs surveyed by the RTK positioning technique is 1.3 cm. The GCP-controlled photogrammetric processing of the UAV images was then successfully performed with an internal precision given by the software as 2.1 cm (horizontal) and 2.8 cm (vertical), respectively. The elevation accuracy of the generated UAV - DEM was then evaluated as $0.2 \pm 6.3$ cm using 167 GNSS RTK points.

Laser spot 6 (weak beam) and Laser spot 5 (strong beam) passed the DEM area and presented 48 ATL06 ice surface points,
among which three were affected by the photons from the CCR and excluded from the validation. The elevation differences between the ATL06 ice surface points and the UAV - DEM were computed and resulted in an estimated ICESat-2 ice surface elevation uncertainty of $1.1 \pm 4.9$ cm.

## 5 Discussions

The validation of ICESat-2 elevations at crossovers between the ground tracks and the CHINARE route using a kinematic
GNSS positioning technique and direct ice surface measurements produced an elevation bias and precision of $1.5 \pm 9.1$ cm for the ATL06 ice surface points, compared to $4.3 \pm 8.5$ cm for the ATL03 photons (Table 2). This indicated that the method used for estimating the ice surface elevations every 20 m along a track in ATL06 data (Smith et al., 2019) achieved a precision that is comparable to that of the original geolocated photons of ATL03 as demonstrated in this study. However, the precision values of 12.5 cm (ATL06) and 11.5 cm (ATL03) in Table B1 for the assessment without the 5 km constraint are relatively larger,
which may be mainly due to non-representativeness of the measured $h_2$ values beyond 5 km and attributed to the local microtopography and uneven firn properties from the coast to the interior of Antarctica along the GNSS traverse. An improvement of the validation system using continuous measurements of $h_2$ along the GNSS traverse is planned for experiments in future expeditions.

The accuracy of the ATL03 data was better validated by using targeted photons returned from CCRs and RTSs. One precisely
GNSS-surveyed CCR was illuminated by a weak beam and showed an offset of 2.1 cm between the GNSS elevation and that of the representative CCR photon. Furthermore, seven precisely GNSS-surveyed check points on an RTS were compared with the corresponding ATL03 photons and presented a bias and precision of $2.5 \pm 2.7$ cm. It should be emphasized that constrained by expedition logistics the observation opportunities of one CCR overpass and seven RTS check points in this study may not be considered as a large sample size. The assessment result may vary with location, environmental conditions, and time. Thus,
there is a need for aggregated opportunities of CCR and RTS observations to achieve a validation result with variant influence factors accounted for (e.g., ATLAS attitude, solar angle, and atmosphere).

The use of the readily available CCRs of 6 cm diameter for the 532 nm wave length of ATLAS, which is larger than 8 mm of the CCRs used in Magruder et al. (2020), is subject to velocity aberration caused by a decreased central disc and receiving signals from the outer lobes of the Fraunhofer diffraction pattern (Born et al., 1999; Magruder et al., 2020; Sun et al., 2019;
Chang et al., 1971). In addition, the larger aperture of the CCR resulted in a higher level of the total signals received by ATLAS so that signals from both the smaller central disc and outer lobes are detected and used in ATL03 data. This may have attributed

to the creation of the long along-track streaks of ~35 m (Fig. 6a) and ~38 m (Fig. 6b) in comparison to those of ~11 m in Magruder et al. (2020). Thus, photons reflected from the lower neighboring CCR that was ~10 m away in the cross-track direction (Fig. 6b) were detected based on the symmetric Fraunhofer diffraction pattern. Similarly, the one-layer photon streak

(green dots in Fig. 6a) may include those reflected from one or both neighboring CCRs because the elevations of all three adjacent CCRs (#4, #5 and #6) are within a 15 cm range (Table C1) due to local ice surface topography and logistic constraints, although the poles were manufactured in different lengths. On the other hand, a temporal distribution of energy within a pulse is approximately Gaussian (Smith et al., 2019); the received signals from the nadir CCR in the central disc are generally of a higher level (about 84% of the total energy) than those from the neighboring CCR in the outer lobes given atmospheric

scattering and other optical losses (Magruder et al., 2020). This increased "signal-to-noise" ratio may have caused presence of the nadir CCR elevations only in the ~9 m central section along the photon streak in Fig. 6b (red rectangle). Therefore, by selecting photons inside the central window of the CCR streak it ensures that high quality photons from the nadir CCR in the central disc of the Fraunhofer diffraction pattern be used to estimate the elevation of the nadir CCR. The result is also validated by the nadir CCR position surveyed by using the high-precision GNSS RTK technique.

The fact that the photons returned from the CCRs are saturated is indicated by the full saturation fraction (FSF) values of 0.429 and 1.000 for the segments containing the illuminated CCRs near Zhongshan Station and Taishan Station, respectively (Table 4). They are drastically higher than those of the ice surface areas nearby (0 and 0.072). It is also noted that the CCR FSF at Zhongshan Station is more than 50% lower than that at Taishan Station (max. FSF). A similar trend exists between FSFs of the ice surface areas at the two sites despite the variation in the ratio. These consistent FSF differences between the two sites

may be explained by their corresponding "total neutral atmospheric delay" (TNAD) in the ATL03 data (*/gtx/geolocation/neutat_delay_total*), which represents the totally neutral atmospheric delay correction (Neumann et al., 2020); TNAD is dependent on the state of the atmosphere, which itself is dependent on the total pressure, partial pressure of water vapor, and air temperature. The TNAD of 2.353 m at Zhongshan Station over that of 1.726 at Taishan Station suggested that atmospheric conditions may have had a severe impact on the CCR photons at Zhongshan Station, including the increased level

of refraction and thus, prolonged the time delay of the returned photons and contributed to less photons reaching the detectors. Therefore, we see less-saturated photons (lower FSF) near Zhongshan Station. And the elevations of the CCR photons at Zhongshan Station are also more dispersed (Fig. 6a) compared to the regularly distributed pattern at Taishan Station (Fig. 6b).

**Table 4. The full saturation fraction (FSF) and total neutral atmospheric delay (TNAD) of the illuminated CCRs near Zhongshan**
**Station and Taishan Station.**

| ICESat-2 beam (site) | FSF of CCR (ratio) | FSF of ice surface (ratio) | TNAD of CCR (m) | TNAD of ice surface (m) |
|---|---|---|---|---|
| **Weak beam** (Zhongshan) | 0.429 | 0 | 2.335 | 2.335 |
| **Weak beam** (Taishan) | 1.000 | 0.072 | 1.726 | 1.726 |

**6 Conclusions**

This paper presents the results of the assessment of ICESat-2 ice surface elevations along the CHINARE route in East Antarctica. The validation campaign was designed and implemented in cooperation with the 36[th] CHINARE Antarctic expedition. An assessment of the ICESat-2 ATL03 and ATL06 data was performed along the 520 km traverse using a kinematic
GNSS positioning technique. Near Zhongshan Station and Taishan Station, additional coordinated multi-sensor observations of a CCR line array, an RTS, and a UAV were acquired for each site. Overall, this systematic validation of the ICESat-2 data covered a variety of the eastern AIS conditions from the coast to inland Antarctica along the 520 km traverse and is complementary to the 750 km traverse validation of flat inland Antarctica along the latitude of 88°S (Brunt et al., 2020b).

The following conclusions are drawn from this research.

1) The comparison of the ICESat-2 Release 003 data with the high-precision GNSS survey and direct snowcat-to-ice surface height measurements along the 520 km CHINARE route showed that the elevations of the ATL06 ice surface points are accurate to 1.5 cm with 9.1 cm precision (1.5 $\pm$9.1 cm), and the elevations of the ATL03 photons are accurate to 4.3 cm with 8.5 cm precision (4.3 $\pm$8.5 cm). This is comparable to the similar result of the ICESat-2 Release 001 data validation,
3 $\pm$9 cm for the ATL06 data and 5 $\pm$13 cm for the ATL03 data along a traverse of the 88°S parallel (Brunt et al., 2019b).

2) The validation of the ICESat-2 ATL03 data using a high-precision GNSS-surveyed CCR showed an accuracy of 2.1 cm. An additional experiment using seven high-precision GNSS-surveyed check points on the RTS near Zhongshan Station indicated that the ATL03 data are accurate to 2.5 cm with 2.7 cm precision.

3) The validation of the ATL06 ice surface points along two ICESat-2 tracks with a high-precision GNSS-controlled UAV -
DEM (750 m $\times$ 500 m) near Zhongshan Station indicated that the ATL06 elevations are accurate to 1.1 cm with 4.9 cm precision.

Overall, our ICESat-2 data validation results show that the elevation of the ATL06 ice surface points is accurate to 1.5 cm with a precision ranging from 4.9 cm in a local UAV - DEM environment near Zhongshan Station to 9.1 cm along the 520 km
CHINARE route. The elevation of the ATL03 photons is accurate to 2.1 cm as estimated by using a line array of CCRs, and accurate to 2.5 cm with 2.7 cm precision as estimated by using RTSs. The validation results demonstrated that the estimated ICESat-2 elevations are accurate to 1.5–2.5 cm in this East Antarctic region, which is higher than prior satellite altimeters and important for eliminating mission biases by overcoming the uncertainties in the estimation of mass balance in East Antarctica. In addition to the different ice surface types covered from coast to inland Antarctica along the 520 km CHINARE route, the
result of the CCR elevation assessment complements that of the CCR horizontal accuracy and footprint size assessment in Magruder et al. (2020). Furthermore, the RTS and UAV – DEM assessment results are reported, to our knowledge, for the first time for the validation of the early stage ICESat-2 data in the AIS environment. Although the UAV - DEM coverage is relatively small and the CCR and RTS observation opportunities are relatively limited in comparison to the large number of

GNSS observations along the 520 km GNSS traverse, their performances and achieved results in this study pave a way for
future applications with aggregated observations at more sites for a more robust assessment. Therefore, our ICESat-2 validation methodology and sensor system will be applied to carry out the continued assessment of the ICESat-2 data, especially for calibration against potential degradation of the elevation measurements during the later operation period.

*Data availability.* The ICESat-2 data were downloaded from the NASA National Snow and Ice Data Center (NSIDC). The
cal/val data used in this study are made available at https://datadryad.org/stash/share/mU52z7OSWAG07tTsGCWXt0Rm0MmT12qwdPORBIUpsnw (last accessed on April 12, 2021).

*Author contributions.* RL led the study and designed the cal/val campaign. TH and GQ carried out field observations. All
authors were involved in data processing, analysis and presentation.

*Competing interests.* The authors declare that they have no conflict of interest.

*Acknowledgements.* We thank editor and three reviewers for their constructive comments and suggestions. The logistic and
technical support provided by the 36th CHINARE team and the Polar Research Institute of China is appreciated. This study was supported by the National Science Foundation of China (41730102), National Key Research and Development Program of China (No. 2017YFA0603100), and Polar Expedition Office of the State Oceanic Administration.

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

## Appendices

### Appendix A. CCR lens

The mechanism of photons entering into and reflecting from a CCR is realized according to the optical total reflection principle (Born et al., 1999). The CCR we used is an optical reflector made of K-3 glass (https://zhongchengyq.1688.com/, last accessed on April 12, 2021) originally designed for laser distance ranging (Fig. A1a). The lens system consists of three adjacent and mutually orthogonal plane-reflecting surfaces, which have an angle of 54.73 °from the horizontal plane (Fig. A1b). An incident beam ① from the satellite comes out of the CCR as the main outgoing beam ② that is parallel to ① but in the opposite direction, allowing it to be received by the satellite. The system guarantees that any photon from the satellite entering the field of view (FOV) of the CCR, ±35 °from the zenith, will be returned to the satellite. Within a 20-meter along - track segment, only one photon, referred to as reference photon, is geolocated in an absolute sense. Other photons are then geolocated with respect to that reference photon (Neumann et al., 2019; Luthcke et al., 2021). Therefore, bounded between the first footprint and last footprint receiving the CCR photons, the elevations of photons returned from the CCR are recorded as a streak of photons in ATL03 data (green dots in Figs. 6a and 6b).

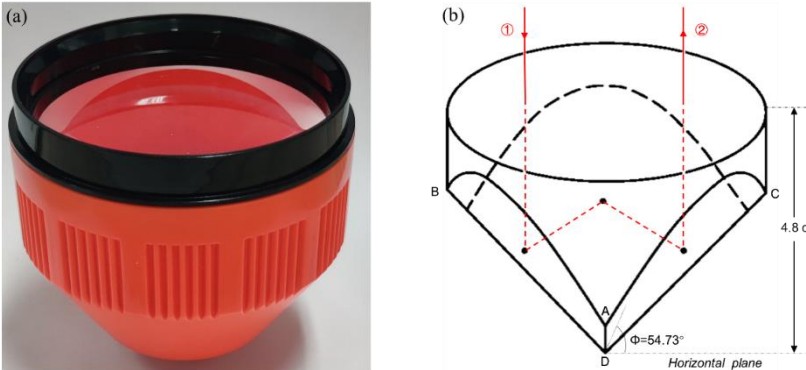

**Figure A1. (a) CCR head used for calibration of ICESat-2 ATL03 data, and (b) parameters of the lens system and optical path of a total-reflecting photon.**

## Appendix B. Assessment using observations at all intersections

**Table B1. Assessment of ICESat-2 ATL06 ice surface points and ATL03 photons using the GNSS PPK technique at all intersections along the GNSS traverse. Bias and precision were estimated from their elevation differences using N ice surface points or photons. The difference is calculated as ICESat-2 elevation minus GNSS elevation.**

| Ground track (Laser spot ID) | ATL06 Bias ± Precision(cm) | ATL03 Bias ± Precision(cm) |
|---|---|---|
| GT1L (Laser spot 6) | +2.9 ±12.0 (N = 111) | +6.4 ±12.6 (N = 2055) |
| GT1R (Laser spot 5) | +2.5 ±12.7 (N = 116) | +4.5 ±12.5 (N = 4542) |
| GT2L (Laser spot 4) | -0.3 ±12.4 (N = 101) | +0.2 ±7.3 (N = 1245) |
| GT2R (Laser spot 3) | -2.8 ±12.4 (N = 106) | +0.5 ±11.8 (N = 4272) |
| GT3L (Laser spot 2) | -1.1 ±12.6 (N = 88) | +2.8 ±10.6 (N = 1233) |
| GT3R (Laser spot 1) | -2.6 ±13.0 (N = 99) | +1.7 ±12.1 (N = 4490) |
| ALL | +0.5 ±12.7 (N = 621) | +3.4 ±11.5 (N = 17839) |

# Appendix C. Elevations of CCRs deployed near Zhongshan Station and Taishan Station

**Table C1. Elevations of CCRs near Zhongshan Station were measured by a GNSS survey at centimeter level accuracy. Heights of CCRs near Taishan Station were estimated from the CCR center offset parameter and distance measured from pole top to ice surface.**

| Site | CCR ID | Latitude | Longitude | Elevation (m) |
|---|---|---|---|---|
| | 1 | 69.43075933°S | 76.28232148°E | 189.853 |
| | 2 | 69.43076510°S | 76.28256045°E | 189.450 |
| | 3 | 69.43077028°S | 76.28280756°E | 190.234 |
| | 4 | 69.43077575°S | 76.28305904°E | 190.444 |
| Zhongshan | 5 | 69.43078215°S | 76.28330064°E | 190.501 |
| Station | 6 | 69.43078586°S | 76.28353504°E | 190.250 |
| | 7 | 69.43079196°S | 76.28376717°E | 189.691 |
| | 8 | 69.43079670°S | 76.28403183°E | 189.876 |
| | 9 | 69.43080191°S | 76.28428890°E | 190.756 |
| | 10 | 69.43080800°S | 76.28453394°E | 190.141 |

| Site | CCR ID | Latitude | Longitude | Height (m) |
|---|---|---|---|---|
| | 1 | 73.78341667°S | 76.98276944°E | 1.425 |
| | 2 | 73.78343611°S | 76.98246111°E | 0.668 |
| | 3 | 73.78345000°S | 76.98213611°E | 0.824 |
| | 4 | 73.78345556°S | 76.98180833°E | 1.605 |
| | 5 | 73.78346944°S | 76.98150278°E | 0.647 |
| Taishan Station | 6 | 73.78348056°S | 76.98116944°E | 1.284 |
| | 7 | 73.78349722°S | 76.98083333°E | 0.841 |
| | 8 | 73.78350278°S | 76.98050833°E | 1.582 |
| | 9 | 73.78351389°S | 76.98024444°E | 0.743 |
| | 10 | 73.78353611°S | 76.97990833°E | 0.651 |