# Peer review of "Assessment of ICESat-2 ice surface elevations over the CHINARE route, East Antarctica, based on coordinated multi-sensor observations"

_The Cryosphere, 2020_

## Referee Comment (RC1) · Anonymous Referee #1 · 14 Dec 2020

Assessment of ICESat-2 ice surface elevations over the CHINARE route, East Antarctica, based on coordinated multi-sensor observations Li et al. The Cryosphere https://doi.org/10.5194/tc-2020-330 8 December 2020

General Reviewer Comments:

This paper presents the results of a validation effort for ICESat-2 ice surface elevations in East Antarctica. The campaign was coordinated with the CHINARE Antarctic expedition from December 2019 to February 2020. The authors focused on an assessment of both the ATL03 (geolocated photons) and ATL06 (land ice elevations) along-track ICESat-2 Level 3a data products. The goal of this effort was to determine the ICESat-2 product vertical accuracy and precision using four different approaches.

The four resources for ICESat-2 validation are:

- 520 km GNSS traverse from the coastal Zhongshan Station to the inland Taishan Station
- Two line arrays of corner cube retro-reflectors
- Two retroreflective target sheets.
- Imagery from two UAV platforms

The authors designed this campaign as a complementary approach to those efforts performed previously by the ICESat-2 mission team and confirm the regional accuracy to support uncertainty determination within mass balance and change rate calculations for East Antarctica. The logistics accomplished in this coordinated approach for validation are commendable and I congratulate the authors on such an effort.

My primary concern with this work is primarily with two of the validation techniques that use ground based retroreflectors. These results are very limited as there are only 7 data points evaluated for the target sheet and 1 data point evaluated for a CCR. Clearly, the traverse comparison provides a robust performance assessment for vertical accuracy based on a reasonable sample size for statistical aggregation. The same does not seem not true for the CCR and RTS approach.

- The authors should include the appropriate data product citations (via NSIDC) in addition to the references to the ATL03 and ATl06 Algorithm Theoretical Basis Documents by Neumann and Smith.
- The Neumann et al., 2020b should be replaced with:

- Magruder, L. Brunt, K., and Alonzo, M. (2020) Early ICESat-2 on-orbit geolocation validation using ground-based corner cube retro-reflectors. *Remote Sensing*, 12(21), 3653; https://doi.org/10.3390/rs12213653
- 'two sets' of CCRs would be better represented as two line arrays. 'Sets' is not applicable to individual CCRs nor does it describe the implementation effectively.
- Is an RTS a 'set' as well?
- L65: CCRs do not 'capture' photons, they reflect them so that the unique signature they provide to ATLAS can be analyzed relative to their known height and position.
- Why do the authors use 6 cm diameter CCRs for ICESat-2 when clearly for 532 nm that size optic would suffer from velocity aberration? Was this considered in the selection? Does that make a difference in the analysis if the returns are coming from the lobes of the diffraction pattern rather than the central disk?
- What are the elevation differences between the 10 CCRs- or what is the combination of elevations in the line array?
- 3.1.3- The ICESat-2 laser pairs are separated by 3.3 km in the across-track direction.
- L195- The authors report a laser diameter value of 11 m in the introduction and then move to estimating a ~13 m. Where does this new value come from?
- Since the weak beam detectors are saturated at ~4 photons how does this contribute to your radiometry study. The ~10 photons for the strong beam is not saturated (which would be 16).
- Figure 6 (c) Should this height be dhCCR? That was described as the distance between the top of the pole and the middle of the CCR, presumably not at the meter scale that is shown here? Also, in this figure why is there a bulge in the CCR signature rather than a straight line as shown in Figure 6 (a)? This seems as if the footprint illuminated multiple CCRs at different heights? This doesn't seem to be normal streak characteristics.
- Section 4.2: The ATL03 CCR streak is 40 pulses across 35 m, how is that with 0.7 cm per pulse. It should be closer to 28 m in length.
- How does the selection of the CCR returns impact the results, meaning if you included more of the streak's length does it negatively impact the comparison or change the quantitative values you are achieving?
- Do you average the elevations per pulse or delta time? There is not a pulse id parameter.
- The authors should emphasize in the paper that the CCR and RTS techniques are localized assessments with results that vary with satellite orientation and time of year (e.g. solar angle). This is particularly important when there are so few opportunities to evaluate the performance (e.g. 1 CCR overpass and 7 RTS points). These techniques cannot yet be compared to those associated with the traverse without further aggregation of opportunities for analysis.
- An additional issue with using just 1 CCR overpass is the impact of the atmosphere on the signal or background noise levels to understand precision. The numbers reported here would represent the variability for one environmental scenario but doesn't capture a representative data quality variability relative to the many influences.

---

## Referee Comment (RC2) · Anonymous Referee #2 · 17 Dec 2020

This manuscript presents the results of a validation of ICESat-2 laser altimetry by kinematic GNSS in East Antarctica. The results of this validation highlight the high quality of both datasets, the satellite data but also the in situ validation data. Together with the validation along the 88°S traverse by Brunt et al., these results provide important insights into the characteristics of the ICESat-2 mission.

The manuscript is well written, the applied methods are appropriate and the results are nicely illustrated. Alone in the structure of the manuscript I would suggest a few changes. It is sometimes difficult to follow a specific method as each section jumps from one dataset to the others. The "Data" section briefly describes the ICESat-2 data

and all the measurements performed during the campaign. Then under "Methods" you describe in detail the GNSS-processing, the height reduction from the antenna to the snow surface, the ICESat-GNSS comparison and the validation with the other measurements. I would suggest to make separate subsections for GNSS, the CCRs, the RTSs and the UAV-DEM and ICESat-2 under "Data" and describe all the details in obtaining the each of these datasets there. Therefore, you could simply move the respective paragraphs from Methods to Data. Then, the "Methods" section could concentrate on the details of the validation between each dataset and ICESat-2.

This little change in the structure would also allow to avoid confusion between the different types of GNSS-processing (which are otherwise easy to be mixed up). If I understand you right, you have 2 types of GNSS observations: a) the GNSS-base stations with permanent observations for ∼3 days (more at Zhongshan) and b) GNSS-rovers on the PistenBullys, the CCR stakes and for the ground control points of the RTS and the UAV-DEM. For the processing of the base stations (a) you use PPP. The coordinates of the rovers (b) then are obtained as differential kinematic coordinates with respect to these base stations. These coordinates are obtained in post processing (if I understand it right), so I wouldn't call that real-time kinematic (RTK). I have several remarks to that GNSS-processing:

1. Similar studies used reference stations at the coast (Schröder et al. 2017, doi: 10.5194/tc-11-1111-2017) or directly processed the rover GNSS-data using PPP (Kohler et al. 2013, doi: 10.1109/TGRS.2012.2207963 or Brunt et al. 2019, doi: 10.5194/tc-13-579-2019). Your processing software (RTKLIB) seems to support these types processing modes. Did you try to process your rovers this way? This would overcome the limited availability of your base station data and even provide useful results for your GNSS-measurements near Taishan.

2. It is quite usual that internal accuracy values reported by the GNSS-software are way to optimistic. You results for internal crossovers in your GNSS-profiles demonstrate that nicely. However, in the vertical GNSS accuracy at the CCR of the ground control points,

you simply state values of ~0.3-0.4 cm without any information about their origin. Is that the accuracy reported by the software? For how long was each point observed? This accuracy is remarkably high for a short observation. Do you have any evidence, that this value is realistic?

3. (this comment refers to l.240) I appreciate that you checked the accuracy of the GNSS-profiles using crossover differences at intersections. However, I would suggest a few more analyses:

- If I understand you correctly, you separate your total profile into shorter sections according to the distance to the base stations and post-process each section with this base station as reference. It would be very interesting to have some overlap in the processing of these the sections. Hence, at the transition from one section to the next one, you could process some measurements with both base station and compare the results.

- Moreover, referring to l.139 you have two antennas. Are they installed on the same PistenBully? If yes, are there any systematic offsets between them and what is the noise? If no, did you check crossovers between the two PistenBullys?

Besides these major point and remarks, I have following detailed comments:

l.21: "...which is important for overcoming the uncertainties in the estimation of mass balance in East Antarctica" is a very general statement. The most important topics in East Antarctic mass balance are probably eliminating mission biases and the conversion from volume to mass. This validation contributes only to the first point.

l.30: remove "As"

l.55-74: I suggest to add brief motivations for each of the methods of validation (kinematic profiles, CCRs, RTSs, UAVs). It would be useful for the reader to know the benefit of each of the methods from the very beginning of the paper.

l.64: Later you describe that you obtain your coordinates in post-processing. So the

survey is not real-time kinematic (RTK). This totally makes sense (as post-processing is much more precise) but please be also precise in describing your method.

Fig.1: In this context, no measurements have been conducted at Great Wall Station. It is fair to mention the original plans in the text but I suggest to remove the inset from Fig.1.

l.95: How is this accuracy obtained? You describe the GNSS-processing and the validation methods later, so such accuracy measures should appear later, when the reader knows how they were obtained. Furthermore, is this really RTK (see comment on l.64)?

l.100: See comments on l.95

l.113: Start a new sentence after "532 nm".

l.134ff: What is the reference frame and the ellipsoid for the GNSS coordinates and the ICESat data? And how about the permanent tide? Many altimetry measurements refer to the Topex ellipsoid and are in the 'mean tide' system while GNSS data generally refer to WGS84 and are 'tide-free'. Please give some details here to show that the different data are comparable.

l.139: Were both antennas mounted on the roof of the same PistenBully?

l.179: The description of the interpolation of h2 fits much better in section 3.1.2.

l.210: I would have expect that the CCR looks like a unique point reflector. Why does it show up as a curve with slightly lower elevations on the sides?

l.226: How was this accuracy obtained? From l.230, I guess that the accuracy of the UAV-DEM is just several meters and you need several ground control points for a more precise absolute orientation. Is this correct? Could you explain this a bit more detailed (or give some references)?

l.264: With regard to the variation between the different ground tracks, the lack of GT2-

results for ATL03 (due to the very strict exclusion conditions) and the precision of ∼9 cm I would be careful concluding that the ATL06 bias is smaller than the ATL03 bias. I don't think that the differences are significant.

l.266ff: You state that the standard deviation of the h2 measurements is ∼3cm and this variation is mainly attributed to the microtopography and firn density changes. However, the effect of microtopography will apply immediately when you move a few meters. Systematic variations in density should be largely accounted for by the IDW interpolation. So in my opinion, these effects alone cannot be responsible for these larger differences when using the full profile. Using only data in a 2km vicinity of the h2 measurements reduces the usable crossovers dramatically. So, before reducing the amount of data so drastically, I suggest to do a few more analysis on these larger difference. Did you do any outlier checks before calculating these standard deviations? A few outliers can have a large effect on stddev. Or is there a specific spatial pattern (maybe in the region around 71°S, which is the largest gap between h2-measurements)?

l.282 Without a precise GNSS-elevation of the CCR, you are comparing ICESat-2 measurements to ICESat-2 measurements of the same orbit. So, all you can do is comparing the offset between photons from the ground and from the CCR to the measured stake height. I would suggest to do this using the ATL03 data only as you cannot be sure, which photons contributed to the mean ATL06 data point.

l.333 As discussed on l.264, I don't believe that the difference in the offset between ALT03 and ATL06 is significant.

---

## Referee Comment (RC3) · Anonymous Referee #3 · 3 Mar 2021

**1  Overview**

Li et al. (2020) deliver results from an Antarctic campaign designed to assess elevation measurements from NASA's Ice Cloud and land Elevation Satellite-2 (ICESat-2). For the most part, this was a well-designed and well-thought-out experiment for evaluating the ICESat-2 data. The work presented by the authors falls within the scope of *The Cryosphere* and could make a good contribution for ICESat-2 calibration and validation (cal/val). Overall, while this is a well-written manuscript, there are a few issues that should be resolved before its publication.

[Figure]

**2 Broad comments**

- The cal/val data from the CHINARE campaign needs to be publicly accessible to be of use in the ICESat-2 project science office and the scientific community

- There are other supporting and cal/val efforts that should be mentioned in the text (e.g. NASA Operation IceBridge, Greenland Summit Station (Brunt et al., 2017), salar de Uyuni (Borsa et al., 2019), and the updated Antarctic pole hole campaign (Brunt et al., 2021))

- The L/R designations of the ICESat-2 beams do not correspond with weak/strong full time as it depends on the orientation of the spacecraft. Might also help to include statistics on laser spots (1–6) to help determine any drift or biases in a given beam.

- While ICESat-2 will presumably help improve mass balance and sea level determination efforts with satellite altimetry, these are not in the mission requirements as they require modeling efforts

- Links included in the text should have labels for when the website was last date accessed. Some of these links can also be simplified by removing optional URL parameters.

**3 Line-by-line comments**

**Page 1, Lines 9–10:** should probably be something like "We present the results of an assessment of ice surface elevation measurements from NASA's Ice Cloud and land Elevation Satellite-2 (ICESat-2) along the CHINARE (CHINese Antarctic Research Expedition) route near the Amery Ice Shelf in East Antarctica."

**Page 1, Line 13:** "...the ICESat-2 geolocated photon product (ATL03) and land ice elevation product (ATL06)..."

**Page 1, Line 17** replace "in a previous study" with " (Brunt et al., 2021)"

**Page 1, Lines 21–22** While this is an important study, it is limited to a small region of East Antarctica covering a small percentage of ICESat-2 reference ground tracks (RGTs). Need to be careful not to overstate the results here. Not sure how these results help overcome the uncertainties in East Antarctic mass balance.

**Page 1, Line 23** What do you mean by "especially during the later operation period"? Do you mean that such field capabilities cannot be implemented for a couple of years, or that it is important to calibrate against potential degradation of the satellite measurements?

**Page 1, Lines 25–27** This sentence is awkwardly phrased.

**Page 2, Line 30** ATLAS is the primary instrument along with the GPS transceivers and the star cameras.

**Page 2, Lines 33-34** ICESat-2 will likely help improve mass balance and sea level contribution estimates from satellite altimetry, but those are not part of the mission requirements.

**Page 2, Line 34** The 0.4 cm/yr target is a mission requirement, not the current state of knowledge or uncertainty in elevation change.

**Page 2, Lines 34–37** This sentence is awkwardly phrased. Could be something like "We use Release-3 of the ICESat-2 geolocated photon elevation (ATL03) and land ice surface elevation (ATL06) products provided by the US National Snow and Ice Data Center (NSIDC)."

**Page 2, Lines 38–39** "The calibration and validation of measurements is important for all satellite missions, particularly for missions with new instruments or technology, such the photon-counting laser altimeter on-board ICESat-2."

**Page 2, Lines 42–43** "Before launch, the ICESat-2 Project Science Office (PSO) funded calibration and validation experiments to be conducted on both the Greenland and Antarctic ice sheets. The annual Antarctic campaigns traverse a 300km stretch of the interior of Antarctica near 88°S covering 20% of the ICESat-2 reference ground tracks (RGTs) (Brunt et al., 2019)."

**Page 2, Line 45** "the Antarctic Ice Sheet (AIS)"

**Page 2, Lines 47–48** "The NASA-led team also placed and used corner cube retroreflectors (CCRs) to collect ICESat-2 signatures at known points to help determine the horizontal geolocation accuracy of the laser pointing determination."

**Page 2, Line 51** Again, while this is an important study, this is not a complete study of the "whole" of Antarctica.

**Page 2, Line 56** Replace "mass" with "volume"

**Page 2, Line 70** horizontal or vertical accuracy?

**Figure 1** Where is the inset map of Great Wall Station located? Does the inset need to be included if the station was not used as part of the campaign?

**Table 1** This is less ICESat-2 data than I would have thought. Are these numbers reduced using quality flags?

**Table 1** What do you mean by "not applicable" for ATL06 geolocation accuracy? That these are simply parameterized in the product?

**Page 5, Line 97** Why were 6 cm CCRs chosen for this campaign?

**Page 5, Line 102** Possible to use a different positioning technique for these to reduce the impact of the station problems? 1 meter vertical is not going to be beneficial from a cal/val standpoint.

**Page 5, Line 109** "that are separated by 90 meters"

**Page 5, Line 115** "We selected a silver-gray coating with R = 0.235 as it was the closest to the reportedly highest estimated probability (EP) of photon detection coating with R = 0.28"

**Page 5, Line 109** Remove "Thus"

**Page 8, Line 167** "which are 3 km apart"

**Page 8, Line 168** Weak/Strong beams can be either left or right depending on the spacecraft orientation.

**Page 8, Line 171** "We reduce the impact of non-signal and noisy measurements by reducing the ATL06 land ice elevation measurements using the *atl06_quality_summary* flag and the ATL03 geolocated photon measurements to medium to high confidence photons using the *signal_conf_ph* flag. We also consider . . . "

**Page 8, Lines 174–175** What determination did you use to decide if buffer to low classified photons should be included? In some cases, the buffer to low classified photons often can improve comparisons with ground measurements due to the shape of the ATLAS transmit pulse (which can be truncated in the ATL03 classifier if only including high-quality PEs).

**Page 8, Line 177** Remove "On the other hand"

**Page 9, Lines 187-189** As this campaign includes multiple different terrains, was slope considered when comparing with the ATL06 measurements? i.e. along-track

slope is estimated when calculating the average surface height in ATL06. Would your ground measurements also provide a metric for the along-track and across-track slopes estimated by ATL06?

**Page 9, Lines 194–196** The early mission pointing issue is a known problem (was due to a reference frame mismatch in the onboard software for the star cameras) (Luthcke et al., 2021).

**Page 10, Lines 206–212** These sentences are awkwardly phrased.

**Page 10, Line 207** 11m laser footprint?

**Page 10, Line 209** "confidence flag "*signal_conf_ph*" equal to middle to high confidence in order to calculate an average elevation in each pulse with reduced noise". You're calculating averages over individual pulses? Are there enough return PEs to calculate at this along-track length with significance? This is fitting at approximately 0.7 meters along track correct?

**Page 10, Line 210** replace "further fitted" with simply "fit".

**Page 10, Line 210** Are you using "e.g." here because you use different curves besides a Gaussian? What other functionals do you consider? Did you mean to use "i.e." here?

**Figure 6** Is there a way of combining the plots for the same region to not repeat information? Maybe at some middle level of zoom?

**Page 11, Line 223** 1.0 cm horizontal?

**Page 12, Line 250** Can you clarify what is 816 meters apart? As phrased it could be interpreted as the GNSS measurements are over 800 meters from your ICESat-2 measurements.

**Table 2** Would be beneficial to have these statistics for laser spots in addition to the oriented beams (to directly map to individual laser beams).

**Page 12, Line 263** Again need to be careful to clarify that L/R do not necessarily map to weak/strong as it depends on the spacecraft orientation.

**Page 13, Line 273** Replace "orbit" with RGT

**Page 13, Line 276** 137 medium to high-quality classified photon returns?

**Page 13, Line 284** 1–2 meters vertical is not going to be accurate enough for cal/val purposes

**Page 13, Lines 286–289** This seems possibly circular to use ICESat-2 to evaluate ICESat-2. What are the uncertainties in ATL06 here? What about horizontal geolocation errors of ATL06 and the CCR impacting the heights? What are the slopes?

**Page 14, Lines 309–310** The weak beam also returns to 1/4th the number of detectors (4 instead of 16) and has different thresholds for saturation in the ATL03 algorithm.

**Page 14, Line 310** This is unfortunate that it was so close to the CCRs.

**Table 3** Should these return photon counts be in count/meter?

**Page 14, Line 321** Replace "orbit" with "RGT"

**Page 15, Lines 341–343** Improving the $h_2$ measurements seems like a good advance for future campaigns.

**Page 17, Line 399** The cal/val data needs to be included here in the *Data availability* section

**Page 21, Line 491–492** I believe that within a 20 meter ATL03 segment, only one photon event is directly geolocated in an absolute sense. The other PEs are then geolocated with respect to that reference PE.

**References**

A. A. Borsa, H. A. Fricker, and K. M. Brunt. A Terrestrial Validation of ICESat Elevation Measurements and Implications for Global Reanalyses. *IEEE Transactions on Geoscience and Remote Sensing*, pages 1–14, 2019. ISSN 0196-2892. doi: `10.1109/TGRS.2019.2909739`.

K. M. Brunt, R. L. Hawley, E. R. Lutz, M. Studinger, J. G. Sonntag, M. A. Hofton, L. C. Andrews, and T. A. Neumann. Assessment of NASA airborne laser altimetry data using ground-based GPS data near Summit Station, Greenland. *The Cryosphere*, 11(2):681–692, 2017. doi: `10.5194/tc-11-681-2017`.

K. M. Brunt, T. A. Neumann, and B. E. Smith. Assessment of ICESat-2 Ice Sheet Surface Heights, Based on Comparisons Over the Interior of the Antarctic Ice Sheet. *Geophysical Research Letters*, 46(22):13072–13078, Nov. 2019. ISSN 0094-8276. doi: `10.1029/2019GL084886`.

K. M. Brunt, B. E. Smith, T. C. Sutterley, N. T. Kurtz, and T. A. Neumann. Comparisons of Satellite and Airborne Altimetry With Ground-Based Data From the Interior of the Antarctic Ice Sheet. *Geophysical Research Letters*, 48(2):2020–90572, Jan. 2021. ISSN 0094-8276. doi: `10.1029/2020GL090572`.

R. Li, H. Li, T. Hao, G. Qiao, H. Cui, Y. He, G. Hai, H. Xie, Y. Cheng, and B. Li. Assessment of ICESat-2 ice surface elevations over the CHINARE route, East Antarctica, based on coordinated multi-sensor observations.li-bib-2021 *The Cryosphere Discussions*, 2020:1–22, 2020. doi: `10.5194/tc-2020-330`.

S. Luthcke, T. Thomas, T. Pennington, T. Rebold, J. Nicholas, D. Rowlands, A. Gardner, and S. Bae. ICESat-2 Pointing Calibration and Geolocation Performance. *Earth and Space Science*, Feb. 2021. ISSN 2333-5084. doi: `10.1029/2020EA001494`.

---

## Author Comment (AC1) · 23 Mar 2021

We thank all three reviewers for their constructive comments and suggestions. This manuscript will be much improved by their input. We have made changes to our manuscript. In the following responses, we use "**bold**" text for reviewer's comments, "non-bold" text for our responses, and "*italic*" for changed text in the manuscript.

**Referee #1**

**General Reviewer Comments:**

**This paper presents the results of a validation effort for ICESat-2 ice surface elevations in East Antarctica. The campaign was coordinated with the CHINARE Antarctic expedition from December 2019 to February 2020. The authors focused on an assessment of both the ATL03 (geolocated photons) and ATL06 (land ice elevations) along-track ICESat-2 Level 3a data products. The goal of this effort was to determine the ICESat-2 product vertical accuracy and precision using four different approaches.**

**The four resources for ICESat-2 validation are:**
- **520 km GNSS traverse from the coastal Zhongshan Station to the inland Taishan Station**
- **Two line arrays of corner cube retro-reflectors**
- **Two retroreflective target sheets.**
- **Imagery from two UAV platforms**

**The authors designed this campaign as a complementary approach to those efforts performed previously by the ICESat-2 mission team and confirm the regional accuracy to support uncertainty determination within mass balance and change rate calculations for East Antarctica. The logistics accomplished in this coordinated approach for validation are commendable and I congratulate the authors on such an effort.**

**My primary concern with this work is primarily with two of the validation techniques that use ground based retroreflectors. These results are very limited as there are only 7 data points evaluated for the target sheet and 1 data point evaluated for a CCR. Clearly, the traverse comparison provides a robust performance assessment for vertical accuracy based on a reasonable sample size for statistical aggregation. The same does not seem not true for the CCR and RTS approach.**

We agree that the GNSS-based traverse comparison provided a robust assessment of the ICESat-2 surface elevations along the CHINARE route, which is complementary to the mission team results along the latitude of 88ºS (Brunt et al., 2020b), considering its number of observations, surface types over the 520 km traverse, and achieved accuracy. Furthermore, the long GNSS traverse, areal UAV tests, and CCR and RTS retroreflectors are designed as a systematic and integrated ICESat-2 assessment approach, although they had different observation opportunities individually during the experiments. In addition to their contributions to the overall assessment approach, the CCR and RTS techniques presented in this paper demonstrate a way of using targeted and controlled photons to assess the ICESat-2 *elevations*, in complement to the result in Magruder et al. (2020) in which the *horizontal accuracy* and footprint size were estimated successfully by

using CCRs. Particularly, the results of these less-costly CCR and RTS devices used in this paper were validated using the high precision GNSS technique and showed a potential for future applications in more sites with different Antarctic conditions and mass balance rates.

Based on your comments and suggestions, we revised the paper to point out a) the need for aggregation of (observation) opportunities in the future, and b) to lower the significance of the CCR and RTS results in comparison to the GNSS – traverse results.

We added the following in **Abstract**.

*"…… It should be emphasized that the results based on the CCR and RTS techniques can be improved by further aggregation of observation opportunities for a more robust assessment……"*

We added the following in the **Discussions** section.

*"…… It should be emphasized that constrained by expedition logistics the observation opportunities of one CCR overpass and seven RTS check points in this study may not be considered as a large sample size. The assessment result may vary with locations, environmental conditions, and times. Thus, there is a need for aggregated opportunities of CCR and RTS observations to achieve a validation result with variant influence factors accounted for (e.g., ATLAS attitude, solar angle, and atmosphere)."*

We added the following in the **Conclusions** section.

*"… … In addition to the different ice surface types covered from coast to inland Antarctica along the 520 km CHINARE route, the result of the CCR elevation assessment complements that of the CCR horizontal accuracy and footprint size assessment in Magruder et al. (2020). Furthermore, the RTS and UAV – DEM assessment results are, to our knowledge, reported first time for the validation of these early stage ICESat-2 data in the AIS environment. Although the UAV - DEM coverage is relatively small and the CCR and RTS observation opportunities are relatively limited in comparison to the large number of GNSS observations along the 520 km GNSS traverse, their performances and achieved results in this study pave a way for future applications with aggregated observations at more sites for a more robust assessment. Therefore, our ICESat-2 validation methodology and sensor system will be applied to carry out the continued assessment of the ICESat-2 data, especially for calibration against potential degradation of the elevation measurements during the later operation period."*

- **The authors should include the appropriate data product citations (via NSIDC) in addition to the references to the ATL03 and ATl06 Algorithm Theoretical Basis Documents by Neumann and Smith.**

We added the data product citation of "NSIDC 2021". It is also stated in "*Data availability*".

- **The Neumann et al., 2020b should be replaced with:**

  o **Magruder, L. Brunt, K., and Alonzo, M. (2020) Early ICESat-2 on-orbit geolocation validation using ground-based corner cube retro-reflectors. Remote Sensing, 12(21), 3653; https://doi.org/10.3390/rs12213653**

We have replaced the citation with the suggested one "Magruder et al., 2020". It is also updated in references.

- **'two sets' of CCRs would be better represented as two line arrays. 'Sets' is not applicable to individual CCRs nor does it describe the implementation effectively.**

We have replaced "*sets*" with "*line arrays*".

- **Is an RTS a 'set' as well?**

Each RTS consists of a set of RTS pieces with two different kinds of coatings. We would like to see if we can keep "sets" for RTSs in this context.

- **L65: CCRs do not 'capture' photons, they reflect them so that the unique signature they provide to ATLAS can be analyzed relative to their known height and position.**

We replaced "*capture*" with "*reflect*".

- **Why do the authors use 6 cm diameter CCRs for ICESat-2 when clearly for 532 nm that size optic would suffer from velocity aberration? Was this considered in the selection? Does that make a difference in the analysis if the returns are coming from the lobes of the diffraction pattern rather than the central disk?**

We add the text in the **Data** section to explain why we use 6 cm diameter CCRs: "*Under the time constraints of equipment shipment before expedition and manufacturing cycle of new products we used ten readily available CCRs of 6 cm diameter (Fig. A1a) at each site, which were designed for ground - based laser distance measurement. They were placed linearly at a 10 m interval across a nominal ICESat-2 ground track to reflect photons from ATLAS (Fig. 2a)……*"

The central disc of our 6 cm aperture CCRs has a diameter of ~10.82 m in comparison to ~40.57 m of the ~8 mm diameter CCRs in Magruder et al. (2020), according to Chang et al. (1971). Here is the Fraunhofer diffraction pattern of our CCR:

$$D = \frac{1.22 \times \lambda \times H}{d} \times 2.$$

[Figure]

**Figure of the Fraunhofer diffraction pattern of the 6 cm diameter CCR (λ=532 nm, H=500 km, and d=6 cm)**

With the larger CCR aperture, in addition to the smaller central disc diameter, the total signal level (for both central disc and lobes) is also increased. Thus, it makes signals from the lobes of the Fraunhofer diffraction

pattern also detected. In the following figure, at an approaching position ($t_{Approaching}$) ATLAS received and accepted signals from lobes of both the nadir CCR #6 (red signal curve) and the neighboring CCR #7 (light blue signal curve), both at a lower signal level; this resulted in the reflected photons of higher elevations (green dots) of CCR #6 and those of lower elevations (green dots) of CCR #7. However, at the nadir CCR position ($t_{Nadir}$) ATLAS received and accepted high level signals from the central disc of the nadir CCR #6 (higher elevation), but may have rejected the lower-level signals from the neighboring CCR #7 (lower elevation) because of the much increased ratio between the signals. This allows us to determine the window size, ~9 m gap of the lower streak, to select photons of the nadir CCR (including those inside the central disc) for CCR elevation estimation.

[Figure]

**Figure for CCR signal analysis at the site near Taishan Station**

We added a paragraph in the **Discussions** section to explain the impact:

"*The use of the readily available CCRs of 6 cm diameter for the 532 nm wave length of ATLAS, which is larger than 8 mm of the CCRs used in Magruder et al. (2020), is subject to velocity aberration caused by a decreased central disc and receiving signals from the outer lobes of the Fraunhofer diffraction pattern (Chang et al., 1971; Born et al., 1999; Sun et al., 2019; Magruder et al., 2020). In addition, the larger aperture of the CCR resulted in a higher level of the total signals received by ATLAS so that signals from both the smaller central disc and outer lobes are detected and used to estimate elevations in ATL03 data. This may have attributed to the creation of the long along-track streaks of ~35 m (Fig. 6a) and ~38 m (Fig. 6c) in comparison to those of ~11 m in Magruder et al. (2020). Thus, photons reflected from the lower neighboring CCR(s) in the cross-track direction (Fig. 6c) were detected for the same reason because of the symmetric Fraunhofer diffraction pattern. Similarly, the one-layer photon streak (green dots in Fig. 6a) may include those reflected from one or both neighboring CCRs because the elevations of all three neighboring CCRs (#4, #5 and #6) are within a 15*

*cm range (Table C1) due to local ice surface topography and logistic constraints, although the poles were manufactured in different lengths. On the other hand, a temporal distribution of energy within a pulse is approximately Gaussian (Smith et al., 2019); the received signals in the central disc are generally of a higher level (about 84% of the total energy) than in the outer lobes given atmospheric scattering and other optical losses (Magruder et al., 2020). Correspondingly, we observe that within the window of the nadir CCR (red rectangle in Fig. 6c) the photons are densely aligned along a curve. The same curve trend appears to continue towards both ends, but diverged by potentially blended signals reflected from neighboring CCRs (Figs. 6a and 6c). Therefore, by selecting photons inside the central window of the CCR streak it ensures that high quality photons in the central disc of the Fraunhofer diffraction pattern be used to estimate the elevation of the representative photon of the nadir CCR through the fitting curve. The result is also validated by the nadir CCR position surveyed by using the high-precision GNSS RTK technique.*"

Fig. 6 is improved to reflect the changes in the text:

[Figure]

**Figure 6. (a) CCR experiment near Zhongshan Station: returned photons (ATL03), GNSS-surveyed CCR position, and ice surface photons (ATL03); (b) elevations averaged in each pulse in the red rectangle in (a) to compare with the GNSS-surveyed position; (c) CCR experiment near Taishan Station: returned photons (ATL03), steel tape-surveyed CCR position, and h_CCR - height between CCR center and ice surface (ATL03); and (d) elevations averaged in each pulse in the red rectangle in (c) to compare with the steel tape-surveyed position.**

We revised and added the text in the **Results** and **Method** sections to explain how we select the appropriate photons to analyze the data:

In the **Results** section "*......Among the CCRs, CCR #5 near Zhongshan Station and CCR #6 near Taishan Station were closest (~0.3 m and ~1.3 m) to the ground tracks and are called nadir CCRs. Given these CCR shifts from ground tracks, uncertainty of the ground tracks themselves of up to ~6.5 m (Magruder et al., 2020), and the CCR interval of ~10 m, we expected to have one to two CCRs falling into a footprint of ~11 m. Fig.*

*6c clearly shows returned photons (green dots) of two layers with an elevation difference of ~50 cm. Furthermore, the photons in the upper layer within a window of ~9 m (red rectangle in Fig. 6c) around the nadir CCR (#6) were received by ATLAS, while those of the lower elevation were not present inside the window. Therefore, we only use the returned photons inside a window of ~9 m around the nadir CCR to validate ICESat-2 elevations*".

In the **Method** section "……*where the photons are generally of high quality and have the confidence tag "signal_conf_ph" equal to 3 or 4 to calculate an average elevation in each pulse to reduce noises and potential atmospheric effect (blue dots in Fig. 6b). The average elevations are then further fitted to a symmetric curve, i.e., a Gaussian curve. The central point (or peak) of the curve is treated as the representative CCR photon position. An offset is then calculated between the positions estimated from the ATL03 photons and the GNSS survey.*"

- **What are the elevation differences between the 10 CCRs or what is the combination of elevations in the line array?**

We added the text in the **Data** section: "*The elevations of the nominal centers of the CCR lens were designed to vary within one meter for identification of individual CCRs from which the signal was reflected. The poles were manufactured before the expedition with different lengths. The actual elevations were influenced by the ice surface topography where they were deployed (Table C1).*" See Table C1 for measured elevations of all CCRs near Zhongshan Station and all CCR heights from the ice surface near Taishan Station.

- **3.1.3 The ICESat-2 laser pairs are separated by 3.3 km in the across-track direction.**

Changed it from 3 km to 3.3 km as suggested. It is changed in Fig. 5a also.

- **L195 The authors report a laser diameter value of 11 m in the introduction and then move to estimating a ~13 m. Where does this new value come from?**

"~13 m" was an early value from website https://icesat-2.gsfc.nasa.gov/science/specs. We changed it now to "~11 m" based on Magruder et al. (2020). We also added this new citation in text.

- **Since the weak beam detectors are saturated at ~4 photons how does this contribute to your radiometry study. The ~10 photons for the strong beam is not saturated (which would be 16).**

Counting the photons nearby CCR and RTS was done within an ice surface segment of 250 m along track. This was performed in two segments on both sides of RTS near Zhongshan and Taishan stations. We found that the selected segments were outside RTS but included CCR. Thus, it resulted in ~4 photons for weak beams and ~10 photons for strong beams. Now we moved the counting segments 300 m away from RTS and CCR. Here are the new counts: "…… *ATL03 data have an average number of photons (confidence flag: low-high) of ~2 for weak beams and ~7 for strong beams per pulse on the ice surface in our study area; ……*" They are not saturated for both weak and strong beams. So, the radiometric characterization of the ice surface is not affected by "saturated" beams. This is also reflected in the radiometric analysis result presented in Tables 3 and 4.

- **Figure 6 (c) Should this height be dh$_{CCR}$? That was described as the distance between the top of the pole and the middle of the CCR, presumably not at the meter scale that is shown here?**

(Fig. 6c): Sorry, it should be h$_{CCR}$ (instead of dh$_{CCR}$) that is the height between CCR center and ice surface (ATL03). It is changed in the figure. So, it is of meter scale.

**Also, in this figure why is there a bulge in the CCR signature rather than a straight line as shown in Figure 6 (a)? This seems as if the footprint illuminated multiple CCRs at different heights? This doesn't seem to be normal streak characteristics.**

We added the text in the **Discussions** section to explain it:

"*...... On the other hand, the received signals in the central disc are generally of a higher level (about 84% of the total energy) than in the outer lobes given atmospheric scattering and other optical losses (Magruder et al., 2020). Correspondingly, we observe that within the window of the nadir CCR (red rectangle in Fig. 6c) the photons are densely aligned along a curve. The same curve trend appears to continue towards both ends, but diverged by potentially blended signals reflected from neighboring CCRs (also see Fig. 6a). Therefore, by selecting photons inside the central window of the CCR streak it ensures that high quality photons in the central disc of the Fraunhofer diffraction pattern be used to estimate the elevation of the representative photon of the nadir CCR through the fitting curve. The result is also validated by the nadir CCR position surveyed by using the high-precision GNSS RTK technique.*"

- **Section4.2: The ATL03 CCR streak is 40 pulses across 35 m, how is that with 0.7 cm per pulse. It should be closer to 28 m in length.**

Sorry, the empty pulses in the ATL03 CCR streak were not counted.

Now we added pulse id range for the streak in Fig. 6a: "*...... The elevations of the photons within each pulse, pulse id (/gtx/heights/ph_id_pulse) from 133 to 145, were averaged and used to fit a Gaussian function curve ......*"

We also added pulse id range for the streak in Fig. 6c: "*...... After that, the photons were averaged within each of pulse (pulse id from 19 to 31, blue dots in Fig. 6d) to fit another Gaussian function ......*"

- **How does the selection of the CCR returns impact the results, meaning if you included more of the streak's length does it negatively impact the comparison or change the quantitative values you are achieving?**

As suggested, we performed an analysis of ΔZ between the ICESat-2 CCR elevations and GNSS surveyed CCR elevations achieved using different streak lengths, including 17 m (a large window size), 13 m (ICESat-2 tech spec: https://icesat-2.gsfc.nasa.gov/science/specs), 11 m (footprint size in Magruder et al., 2020), and 9 m (this study).

| Site \ ΔZ (W-size) | ΔZ±σ for 17 m (cm) | ΔZ±σ for 13 m (cm) | ΔZ±σ for 11 m (cm) | ΔZ±σ for 9 m (cm) |
|---|---|---|---|---|
| Zhongshan Station | 2.0±7.7 | 2.4±5.1 | 1.4±3.8 | 2.1±3.7 |
| Taishan Station | 32.3±4.4 | 34.9±3.1 | 35.8±2.5 | 36.0±1.6 |

The CCR position near Taishan Station has an accuracy of ~1 m and the result may not be very meaningful. On the other hand, the CCR position near Zhongshan Station was surveyed by GNSS at a centimeter level accuracy. The elevation differences vary within 1 cm. Considering both elevation difference and fitting error, the smaller window sizes (9 m and 11 m) have better results. The errors with the larger windows may be attributed to the lower quality photons from the outer lobes and possibly reflected from multiple CCRs. We suggest that the window size be limited to between 9 m to 11 m (within one footprint).

- **Do you average the elevations per pulse or delta time? There is not a pulse id parameter.**

Yes, the elevations are averaged per pulse.

We added pulse ids for the streak in Fig. 6a: "…… *The elevations of the photons within each pulse, pulse id (/gtx/heights/ph_id_pulse) from 133 to 145, were averaged and used to fit a Gaussian function curve ……*"

We also added pulse ids for the streak in Fig. 6c: "…… *After that, the photons were averaged within each of pulse (pulse id from 19 to 31, blue dots in Fig. 6d) to fit another Gaussian function ……*"

- **The authors should emphasize in the paper that the CCR and RTS techniques are localized assessments with results that vary with satellite orientation and time of year (e.g. solar angle). This is particularly important when there are so few opportunities to evaluate the performance (e.g. 1 CCR overpass and 7 RTS points). These techniques cannot yet be compared to those associated with the traverse without further aggregation of opportunities for analysis.**

We accepted the suggestion and added texts in three places.

We added the following in **Abstract**.

"…… *It should be emphasized that the results based on the CCR and RTS techniques can be improved by further aggregation of observation opportunities for a more robust assessment ……*"

We added the following in the **Discussions** section.

"…… *It should be emphasized that constrained by expedition logistics the observation opportunities of one CCR overpass and seven RTS check points in this study may not be considered as a large sample size. The assessment result may vary with locations, environmental conditions, and times. Thus, there is a need for aggregated opportunities of CCR and RTS observations to achieve a validation result with variant influence factors accounted for (e.g., ATLAS attitude, solar angle, and atmosphere).*"

We added the following in the **Conclusions** section.

"… … *In addition to the different ice surface types covered from coast to inland Antarctica along the 520 km CHINARE route, the result of the CCR elevation assessment complements that of the CCR horizontal accuracy*

*and footprint size assessment in Magruder et al. (2020). Furthermore, the RTS and UAV – DEM assessment results are, to our knowledge, reported first time for the validation of these early stage ICESat-2 data in the AIS environment. Although the UAV - DEM coverage is relatively small and the CCR and RTS observation opportunities are relatively limited in comparison to the large number of GNSS observations along the 520 km GNSS traverse, their performances and achieved results in this study pave a way for future applications with aggregated observations at more sites for a more robust assessment. Therefore, our ICESat-2 validation methodology and sensor system will be improved to carry out the continued assessment of the ICESat-2 data, especially ATLAS performance during its later operation period.*"

- **An additional issue with using just 1 CCR overpass is the impact of the atmosphere on the signal or background noise levels to understand precision. The numbers reported here would represent the variability for one environmental scenario but doesn't capture a representative data quality variability relative to the many influences.**

We also accepted this suggestion and the changes are made in three sections as seen above.

---

## Author Comment (AC2) · 23 Mar 2021

We thank all three reviewers for their constructive comments and suggestions. This manuscript will be much improved by their input. We have made changes to our manuscript. In the following responses, we use "**bold**" text for reviewer's comments, "non-bold" text for our responses, and "*italic*" for changed text in the manuscript.

**Referee #2**

**This manuscript presents the results of a validation of ICESat-2 laser altimetry by kinematic GNSS in East Antarctica. The results of this validation highlight the high quality of both datasets, the satellite data but also the in situ validation data. Together with the validation along the 88S traverse by Brunt et al., these results provide important insights into the characteristics of the ICESat-2 mission.**

**The manuscript is well written, the applied methods are appropriate and the results are nicely illustrated. Alone in the structure of the manuscript I would suggest a few changes. It is sometimes difficult to follow a specific method as each section jumps from one dataset to the others. The "Data" section briefly describes the ICESat-2 data and all the measurements performed during the campaign. Then under "Methods" you describe in detail the GNSS-processing, the height reduction from the antenna to the snow surface, the ICESat-GNSS comparison and the validation with the other measurements. I would suggest to make separate subsections for GNSS, the CCRs, the RTSs and the UAV-DEM and ICESat-2 under "Data" and describe all the details in obtaining the each of these datasets there. Therefore, you could simply move the respective paragraphs from Methods to Data. Then, the "Methods" section could concentrate on the details of the validation between each dataset and ICESat-2.**

(**Data** and **Method** sections) We restructured the **Data** and **Method** sections as suggested. We separated the **Data** section into subsections of **ICESat-2 data, GNSS data, CCR data, RTS data** and **UAV data**. We also moved the data acquisition related paragraphs from the **Method** section to the **Data** section, as suggested.

**This little change in the structure would also allow to avoid confusion between the different types of GNSS-processing (which are otherwise easy to be mixed up). If I understand you right, you have 2 types of GNSS observations: a) the GNSS-base stations with permanent observations for ~3 days (more at Zhongshan) and b) GNSS rovers on the PistenBullys, the CCR stakes and for the ground control points of the RTS and the UAV-DEM. For the processing of the base stations (a) you use PPP. The coordinates of the rovers (b) then are obtained as differential kinematic coordinates with respect to these base stations. These coordinates are obtained in post processing (if I understand it right), so I wouldn't call that real-time kinematic (RTK). I have several remarks to that GNSS-processing:**

After restructuring we put all GNSS data processing techniques into one place under **3.1 GNSS data processing** in the **Method** section where we introduced:
a) PPP: post processing for base stations;
b) PPK: post processing for receivers on PistenBully along the CHINARE route; it was called RTK in the last version, now it is corrected to PPK; and

c) RTK: real-time positioning of CCRs, RTSs and ground control points for UAV - DEM near Zhongshan Station.

**1. Similar studies used reference stations at the coast (Schröder et al. 2017, doi: 10.5194/tc-11-1111-2017) or directly processed the rover GNSS-data using PPP (Kohler et al. 2013, doi: 10.1109/TGRS.2012.2207963 or Brunt et al. 2019, doi:10.5194/tc-13-579-2019). Your processing software (RTKLIB) seems to support these types processing modes. Did you try to process your rovers this way? This would overcome the limited availability of your base station data and even provide useful results for your GNSS-measurements near Taishan.**

As described in the paper, PPP worked well for post processing using base stations where observations lasted up to ~3 days at each station. And PPK worked well for post processing at positions of the snowcat in motion.

We did try to process our snowcat GNSS data using the PPP technique, in addition to the PPK results. Since the quality of the PPP results are lower (see following table) the number of calculated crossovers is reduced from 26 to 21. So the PPP – PPK comparison is based on 21 crossovers. The accuracy estimated from the elevation differences at the crossovers are

| Solution | Inbound | | Outbound | |
|---|---|---|---|---|
| | Ave. (cm) | Std. (cm) | Ave. (cm) | Std. (cm) |
| PPK | 0.2 | 5.5 | -3.4 | 8.7 |
| PPP | -3.9 | 13.4 | -16.2 | 18.4 |

PPK performed better than PPP in our case. Thus, we used the PPK results. Thanks for this question regarding overcoming the limited availability of base stations in the AIS environment. We will take this suggestion into account if we would conduct another campaign in the future.

Due to logistic difficulties, GNSS-measurements near Taishan Station were carried out using the single-point positioning technique. The GNSS function is integrated into a field surveying pad system. The data are not appropriate to support a PPP solution. Similarly, we will also look into a potential PPP application for these single point situations in the future.

**2. It is quite usual that internal accuracy values reported by the GNSS-software are way to optimistic. You results for internal crossovers in your GNSS-profiles demonstrate that nicely. However, in the vertical GNSS accuracy at the CCR of the ground control points, you simply state values of ~0.3-0.4 cm without any information about their origin. Is that the accuracy reported by the software? For how long was each point observed? This accuracy is remarkably high for a short observation. Do you have any evidence, that this value is realistic?**

In the **GNSS data** subsection we added text to address observation time "……*10 CCRs, 137 randomly distributed GNSS points on the RTS, and 3 GCPs for UAV geometric control near Zhongshan Station were surveyed using the RTK technique. The observation time at each point was about 4 to 5 seconds …...*"

In the **Method** section under **3.1 GNSS data processing** we added a paragraph to address where the internal precisions are from:

"*The RTK positioning technique is applied to estimate positions of the CCRs and GNSS points on the RTS sheets near Zhongshan Station. We used a known GNSS control point at Zhongshan Station as a reference point for RTK. The CCR and RTS checkpoint positions were estimated in real-time by the GNSS receiver's onboard software (https://www.chcnav.com/uploads/i70_DS_EN.pdf). Additionally, the UAV - DEM reconstruction was geometrically controlled using 3 GNSS GCPs. The positions were estimated by the RTK technique implemented in the UAV package (https://www.dji.com/hk-en/phantom-4-rtk?site=brandsite&from=nav). The accuracy of the RTK positions is estimated based on internal precisions given by the applied GNSS systems and the accuracy of the reference point.*"

The GNSS system reported internal precisions of individual CCRs, from which an overall RTK internal precision (optimistic) is estimated as 0.3 cm (horizontal) and 0.4 cm (vertical), respectively. Sorry, we did not account for the GNSS reference point error in the last version of the paper. The GNSS reference point was surveyed four times (~7-10 hours each time). The elevations are (-75.9516 m, -75.9617 m, -75.9553 m, -75.9286 m) with $1\sigma$ of 1.2 cm. Thus, the overall accuracy of CCRs is 1.3 cm.

In the **Results** section we revised the text to give external accuracy "……*Around 6 h before the ICESat-2 pass, the CCRs were deployed and RTK GNSS was surveyed. Based on the internal precisions of individual CCRs given by the GNSS system and accuracy of the known GNSS reference point, the elevation accuracy of ten CCRs is 1.3 cm.…..*"

**3. (this comment refers to l.240) I appreciate that you checked the accuracy of the GNSS-profiles using crossover differences at intersections. However, I would suggest a few more analyses:**

**- If I understand you correctly, you separate your total profile into shorter sections according to the distance to the base stations and post-process each section with this base station as reference. It would be very interesting to have some overlap in the processing of these the sections. Hence, at the transition from one section to the next one, you could process some measurements with both base station and compare the results.**

Thank you for this interesting suggestion. Constrained by our familiarity to the software system, available time, and level of our understanding of your suggestion, we may not have implemented it in the exact way you want. However, we may have got it close. From each base station we extended each segment from ~100 km to ~200 km (red arrowed lines in the following figure), along which we used PPK technique to solve for rover positions (use only one base station, instead of two).

[Figure]

We plot the internal precision profiles of each ~200 km inbound segments in the following figure (colors do not have a specific meaning). They show that there seems to have a linear trend of decreased precision as distance from a base station increases. The extended segments from ~100 km to ~200 km generally have greater vertical errors than the first half. Overall, the profiles of the flat inland interior (beyond ~100 km from Zhongshan Station) showed better performance (particularly in their first ~100 km).

[Figure]

We then moved each profile to start from the same origin in the following figure to examine the internal precision trend vs. distance controlled by a single base station. The linear trend is clear. Thus, we use a linear function $y = k \cdot x + b$ to model the relationship between the internal precision $y$ and distance from coast $x$. We have three cases: Traverse (0 km – 200 km), Segment 1 (0 km – 100 km), and Segment 2 (100 km – 200 km). The average $\pm$ standard deviation of the internal precisions for the three cases are 1.6 $\pm$ 0.3 cm, 1.4 $\pm$ 0.2 cm, and 1.9 $\pm$ 0.2 cm, respectively.

[Figure]

Within the first ~100 km (Segment 1) errors increase at a steeper rate of 0.6 cm per 100 km and reach up to

~1.7 cm; this may be mainly attributed to the effect on the vehicle by rugged terrain and topographic change from coast to inland. The rate of the second half (Segment 2) is lower at 0.2 cm per 100 km where the ice surface is flat.

| Segment | k (cm / 100 km) | b (cm) |
|---------|-----------------|--------|
| Traverse | 0.5 | 1.2 |
| Segment 1 | 0.6 | 1.1 |
| Segment 2 | 0.2 | 1.7 |

On the other hand, we also use elevation differences at crossovers to assess the accuracy before and after the extension of the controlled distance of the GNSS base stations. The extension of the segments from ~100 km to ~200 km resulted in a non-significant uncertainty increase from $0.6 \pm 5.4$ cm to $1.2 \pm 5.9$ cm.

In summary, based on the above analysis the extension of the segments from ~100 km to ~200 km does not cause a significant increase of the internal and external uncertainties. Since this work is still being optimized and the detailed analysis will be presented in a dissertation, we hope that you would agree that we will not put the detailed result in this paper. However, it should guide us in designing our future work in potentially another expedition.

**- Moreover, referring to l.139 you have two antennas. Are they installed on the same PistenBully? If yes, are there any systematic offsets between them and what is the noise? If no, did you check crossovers between the two PistenBullys?**

Antenna 1 and Antenna 2 are installed on the roof of Pistenbully Polar300 (one vehicle) with a fixed offset (24.3 cm as measured by a steel tape).

[Figure]

Top view

The configuration was designed to have a check of GNSS observations for the hard baseline. However, as stated now in the **2.2 GNSS Data** subsection: "*…… Antenna 2 served an ice penetrating radar equipment during the inbound trip. Due to inter-equipment interferences and incidental battery problems, the GNSS rover surveying was carried out by a combination of two receivers……*", the data rate for Antenna 2 was set as 10 Hz to match that of the radar device, compared 1 Hz for Antenna 1. That caused drifted positions of Antenna 2 with an uncertainty of $-8.3 \pm 10.0$ cm as assessed with crossovers, compared to $0.2 \pm 5.5$ cm for Antenna 1.

We consulted with the venders and they confirmed the potential interferences between the devices, which they have also encountered before with the same models of devices. The batteries had a few incidental problems. Thus, we ended up with covering the entire traverse with only one antenna at any time (inbound with Antenna 1; first ~80 km of outbound with Antenna 1, rest of outbound with Antenna 2 when the radar device was off), and being unable to use two antennas to do a reasonable check of the hard baseline.

**Besides these major point and remarks, I have following detailed comments:**

**l.21: "...which is important for overcoming the uncertainties in the estimation of mass balance in East Antarctica" is a very general statement. The most important topics in East Antarctic mass balance are probably eliminating mission biases and the conversion from volume to mass. This validation contributes only to the first point.**

The sentence is revised: "…… *which is important for eliminating mission biases by overcoming the uncertainties in the estimation of mass balance in East Antarctica…...*"

**l.30: remove "As"**

Removed.

**l.55-74: I suggest to add brief motivations for each of the methods of validation (kinematic profiles, CCRs, RTSs, UAVs). It would be useful for the reader to know the benefit of each of the methods from the very beginning of the paper.**

We changed it to a separate paragraph to serve this purpose:
"*In order to validate the ATL03 and ATL06 data along the CHINARE route from the coastal Zhongshan Station to the inland Taishan Station, two roving GNSS receivers of CHC i70 from CHC Navigation Technology LTD (http://www.huace.cn/product/product_show/291) were installed on roof of a snowcat, Pisten Bully Polar 300, to measure ice surface elevations using the post processed kinematic (PPK) positioning technique. Supported by the precise point positioning (PPP) technique, five GNSS base stations with CHC i70 receivers were deployed every ~100 km along the traverse to enable the PPK positioning of the vehicle. Two line arrays of ten upward-looking CCRs (optical prisms) with known elevations were deployed at sites near Zhongshan Station and Taishan Station, respectively, to reflect photons for the verification of individual photons. We used one rectangular (5 m × 150 m) RTS for each site to investigate the reflectivity and elevation accuracy of photons reflected from selected RTS coatings. Finally, two UAVs, DJI Phantom 4 (https://www.dji.com/hk-en/phantom-4-rtk?site=brandsite&from=nav), were used to acquire images for the generation of digital elevation models (DEMs) for an areal assessment of ICESat-2 elevation accuracy. The real-time kinematic (RTK) positioning technique was applied to provide horizontal and vertical positions of the CCRs, GNSS points on RTSs, and control points for UAV - DEM reconstruction…...*"

**l.64: Later you describe that you obtain your coordinates in post-processing. So the survey is not real-time kinematic (RTK). This totally makes sense (as post-processing is much more precise) but please be also precise in describing your method.**

They are corrected. We have changed "RTK" to "PPK" in related places in **Introduction** and elsewhere.

**Fig.1: In this context, no measurements have been conducted at Great Wall Station. It is fair to mention the original plans in the text but I suggest to remove the inset from Fig.1.**

(Figure 1) The Great Wall Station is now removed in the inset.

[Figure]

**l.95: How is this accuracy obtained? You describe the GNSS-processing and the validation methods later, so such accuracy measures should appear later, when the reader knows how they were obtained. Furthermore, is this really RTK (see comment on l.64)?**

As you suggested, the **Data** section is restructured. This part is now under GNSS data subsection.

(Also see above response "Line 64") It should be "PPK". We have changed "RTK" to "PPK".

Now we clarify the positioning techniques of PPP, PPK and RTK in the **Introduction** section.

The description of the PPK accuracy is given in the **Method** section:

"*The internal precisions of the estimated positions from the PPP and PPK processing are given by the software systems. Furthermore, we used the accuracy computed from elevation differences at crossovers where the GNSS surveyed track intersected itself, as shown in Fig. 2c. These crossovers are the intersections of tracks by the snowcat during instrument installations, observations, and overnight breaks. Within a neighborhood of the intersection, we fit two lines to compute the crossover location and elevation difference (Kohler et al., 2013).*"

We moved the estimated PPK accuracy from the **Data** section to the **Results** section:

"*...... which have an average internal elevation precision of 1.6 ± 0.6 cm given by the software system. Finally, the elevation accuracy of the GNSS traverse was assessed as 0.3 ± 5.8 cm by using 26 crossovers of the traverse itself (Fig. 2c) where the GNSS surveyed elevations from two intersecting traverse segments were*

*compared.*"

**l.100: See comments on l.95**

The general description of the RTK accuracy is given in the **Method** section:

 "*...... The accuracy of the RTK positions is estimated based on internal precisions given by the applied GNSS receivers and the accuracy of the reference point.*"

The estimated RTK accuracy for CCRs and RTSs are now given in the **Results** section: "*...... Based on the internal precisions of individual CCRs given by the GNSS system and accuracy of the known GNSS reference point, the elevation accuracy of ten CCRs is 1.3 cm......*"

**l.113: Start a new sentence after "532 nm".**

It is changed accordingly.

**l.134: What is the reference frame and the ellipsoid for the GNSS coordinates and the ICESat data? And how about the permanent tide? Many altimetry measurements refer to the Topex ellipsoid and are in the 'mean tide' system while GNSS data generally refer to WGS84 and are 'tide-free'. Please give some details here to show that the different data are comparable.**

We added a paragraph in the **Method** section to clarify the reference frames:
"*In the ICESat-2 products geographic coordinates (latitude and longitude) are defined based on the WGS84 ellipsoid and heights are referenced to the ITRF2014 frame (Brunt et al., 2019; Neumann et al., 2019); corrections for solid earth tides, ocean loading, solid earth pole tide, ocean pole tide and others are applied to the ATL03 data (Neumann et al., 2019). On the other hand, the processed GNSS data are also referenced based on the WGS84 ellipsoid (Schröder et al., 2017); the ITRF2014 reference frame is used in the GFZ precise ephemeris and precise orbit products which is input to the RTKLIB and MUSIP post processing systems; furthermore, the geophysical corrections for the above tides are applied (Petit and Luzum, 2010). Thus, the reduced ice surface heights are "tide-free" and the permanent crustal deformation is removed (Schröder et al., 2017; Brunt et al., 2021).*"

The cited papers are added to **References**:

**l.139: Were both antennas mounted on the roof of the same PistenBully?**

Yes, please see the above response to "**3. / – Moreover ......**"

**l.179: The description of the interpolation of h2 fits much better in section 3.1.2.**

As suggested, we moved the sentence to section 3.2.1.

**l.210: I would have expect that the CCR looks like a unique point reflector. Why does it show up as a curve with slightly lower elevations on the sides?**

The photon streak length is a function of the CCR diameter, among other parameters. The mission team used an 8 mm diameter CCR that resulted in a flat streak of ~11 m (Magruder et al., 2020). We used a 6 cm diameter

CCR and obtained a ~38 m near Taishan Station (and ~34 m near Zhongshan Station) curved streak.

The following is extracted from the responses to Reviewer #1's comments.
The central disc of our 6 cm aperture CCRs has a diameter of ~10.82 m in comparison to ~40.57 m of the ~8 mm diameter CCRs in Magruder et al. (2020), according to Chang et al. (1971). Here is the Fraunhofer diffraction pattern of our CCR:

$$D = \frac{1.22 \times \lambda \times H}{d} \times 2.$$

[Figure]

**Figure of the Fraunhofer diffraction pattern of the 6 cm diameter CCR (λ=532 nm, H=500 km, and d=6 cm)**

With the larger CCR aperture, in addition to the smaller central disc diameter, the total signal level (for both central disc and lobes) is also increased. Thus, it makes signals from the lobes of the Fraunhofer diffraction pattern also detected. In the following figure, at an approaching position ($t_{Approaching}$) ATLAS received and accepted signals from lobes of both the nadir CCR #6 (red signal curve) and the neighboring CCR #7 (light blue signal curve), both at a lower signal level; this resulted in the reflected photons of higher elevations (green dots) of CCR #6 and those of lower elevations (green dots) of CCR #7. However, at the nadir CCR position ($t_{Nadir}$) ATLAS received and accepted high level signals from the central disc of the nadir CCR #6 (higher elevation), but may have rejected the lower level signals from the neighboring CCR #7 (lower elevation) because of the much increased ratio between the signals. This allows us to determine the window size, ~9 m gap of the lower streak, to select photons of the nadir CCR (including those inside the central disc) for CCR elevation estimation.

[Figure]

**Figure for CCR signal analysis at the site near Taishan Station**

We added a paragraph in the **Discussions** section to explain the impact:

"*The use of the readily available CCRs of 6 cm diameter for the 532 nm wave length of ATLAS, which is larger than 8 mm of the CCRs used in Magruder et al. (2020), is subject to velocity aberration caused by a decreased central disc and receiving signals from the outer lobes of the Fraunhofer diffraction pattern (Chang et al., 1971; Born et al., 1999; Sun et al., 2019; Magruder et al., 2020). In addition, the larger aperture of the CCR resulted in a higher level of the total signals received by ATLAS so that signals from both the smaller central disc and outer lobes are detected and used to estimate elevations in ATL03 data. This may have attributed to the creation of the long along-track streaks of ~35 m (Fig. 6a) and ~38 m (Fig. 6c) in comparison to those of ~11 m in Magruder et al. (2020). Thus, photons reflected from the lower neighboring CCR(s) in the cross-track direction (Fig. 6c) were detected for the same reason. Similarly, the one-layer photon streak (green dots in Fig. 6a) may include those reflected from one or both neighboring CCRs because the elevations of all three CCRs (#4, #5 and #6) are within a 15 cm range (Table A1) due to local ice surface topography and logistic constraints, although the poles were manufactured in different lengths. On the other hand, the received signals in the central disc are generally higher (about 84% of the total energy) than in the outer lobes given atmospheric scattering and other optical losses (Magruder et al., 2020). Correspondingly, we observe that within the window of the nadir CCR (red rectangle in Fig. 6c) the photons are densely aligned along a curve. The same curve trend appears to continue towards both ends, but diverged by potentially blended signals reflected from neighboring CCRs (Figs. 6a and 6c). Therefore, by selecting photons inside the central window of the CCR streak it ensures that high quality photons in the central disc of the Fraunhofer diffraction pattern be used to estimate the elevation of the representative photon of the nadir CCR through the fitting curve. The result is also validated by the nadir CCR position surveyed by using the high-precision GNSS RTK technique.*"

[Figure]

**Figure 6.** (a) CCR experiment near Zhongshan Station: returned photons (ATL03), GNSS-surveyed CCR position, and ice surface photons (ATL03); (b) elevations averaged in each pulse in the red rectangle in (a) to compare with the GNSS-surveyed position; (c) CCR experiment near Taishan Station: returned photons (ATL03), steel tape-surveyed CCR position, and h$_{CCR}$ - height between CCR center and ice surface (ATL03); and (d) elevations averaged in each pulse in the red rectangle in (c) to compare with the steel tape-surveyed position.

**l.226: How was this accuracy obtained? From l.230, I guess that the accuracy of the UAV-DEM is just several meters and you need several ground control points for a more precise absolute orientation. Is this correct? Could you explain this a bit more detailed (or give some references)?**

We revised the paragraph in the **Method** section to clarify the photogrammetric process. We also gave a reference for photogrammetric orientations.

*"In the mapping area a set of ground control points (GCPs) are surveyed using the RTK positioning technique. They are further used to perform a photogrammetric absolute orientation (McGlone, 2013). The 3D surface points are reconstructed from the UAV images by using the structure-from-motion multi-view stereo (SfM-MVS) algorithm (James and Robson, 2012; Turner et al., 2014) implemented in the Pix4Dmapper software (version 4.5.6, https://support.pix4d.com/hc/en-us/categories/360001503192-Pix4Dmapper). As the result, a UAV-DEM and an orthophoto at a centimeter level accuracy (both horizontal and vertical) are generated. Thereafter, we evaluate the elevation differences $\Delta H$ between the elevations of the ICESat-2 ATL06 ice surface points ($H_{Ice\ surface}$) and the corresponding elevations of the UAV-DEM ($H_{UAV\_DEM}$):*

$$\Delta H = H_{Ice\ surface} - H_{UAV\_DEM}. \hspace{3cm} (3)"$$

In the **Results** section we present the estimated accuracy:

*".......A GCP-controlled photogrammetric processing of the UAV images was successfully performed with an internal precision given by the software as 2.1 cm (horizontal) and 2.8 cm (vertical), respectively. The*

*elevation accuracy of the generated UAV-DEM was then evaluated as 0.2 ± 6.3 cm using 167 GNSS RTK points. On the other hand, the 1L (weak beam) and 1R (strong beam) tracks have 48 ATL06 ice surface points in the DEM area, among which three were affected by the photons from the CCR and excluded from the validation. The elevation differences between the ATL06 ice surface points and the UAV-DEM were computed and resulted in an estimated ICESat-2 ice surface elevation uncertainty of 1.1 ± 4.9 cm.*"

**l.264: With regard to the variation between the different ground tracks, the lack of GT2-results for ATL03 (due to the very strict exclusion conditions) and the precision of ~9 cm I would be careful concluding that the ATL06 bias is smaller than the ATL03 bias. I don't think that the differences are significant.**

We agree with you and changed the text accordingly. Please see the response to **Comment l.333** (last page).

**l.266ff: You state that the standard deviation of the h2 measurements is ~3cm and this variation is mainly attributed to the microtopography and firn density changes. However, the effect of microtopography will apply immediately when you move a few meters. Systematic variations in density should be largely accounted for by the IDW interpolation. So in my opinion, these effects alone cannot be responsible for these larger differences when using the full profile. Using only data in a 2 km vicinity of the h2 measurements reduces the usable crossovers dramatically. So, before reducing the amount of data so drastically, I suggest to do a few more analysis on these larger difference. Did you do any outlier checks before calculating these standard deviations? A few outliers can have a large effect on stddev. Or is there a specific spatial pattern (maybe in the region around 71S, which is the largest gap between h2-measurements)?**

That is a good suggestion, thanks. The following plot of ATL06 – GNSS elevation differences (red and blue dots) does show that there is a segment of traverse in the red rectangle where outliers occurred. So, we checked the field measurement notes that show that the closest three $h_2$ measurements (diamonds on the bottom) were erroneously measured to the bottom of wheel-chain prints, while all other $h_2$ values were measured to the ice surface. They were the first three ice surface measurements of the traverse.

[Figure]

**Figure of elevation differences at crossovers between ICESat-2 (ATL06) tracks and the GNSS traverse with red dots for inbound and blue dots for outbound pairs, and diamonds for $h_2$ measurement locations.**

We performed the following experiments to test the impact of the outliers: a) including just each one of the three $h_2$ outliers (along with rest), and b) including all three $h_2$ outliers (along with rest; this had the lower precision result in the appendix of the previous manuscript). No distance limit to $h_2$ locations is applied. All results of Experiment a) are similar to that of Case b). Thus, we had to give up all three $h_2$ measurements (outliers), correspondingly also the ATL06 – GNSS pairs in the red rectangle in the above figure since they are now greater than ~65 km away from the closet $h_2$ measurement.

In the further analysis, we have two sets of results: Table 2. Results with a limit of ~5 km distance from $h_2$ measurements, and Table B1. Results with all intersections along the traverse without limit of distance.

[revised manuscript text omitted]

**l.282 Without a precise GNSS-elevation of the CCR, you are comparing ICESat-2 measurements to ICESat-2 measurements of the same orbit. So, all you can do is comparing the offset between photons from the ground and from the CCR to the measured stake height. I would suggest to do this using the ATL03 data only as you cannot be sure, which photons contributed to the mean ATL06 data point.**

As suggested, we modified Figure 6 (a) and (c) (see Fig. 6 above). We recalculated the CCR elevation offset using the ATL03 data along with the stake height. The text is revised:

"*CCR #6 was found to have returned 52 photons in 13 pulses from the weak beam track (2L) within the ~9 m window in Fig. 6c. To estimate a CCR #6 elevation that is more accurate than the meter-level GNSS result, we first used the ATL03 ice surface photons (black dots in Fig. 6c) to fit a terrain surface plane with an $R^2$ of 0.9959. Using the measured CCR height $h_{CCR}$ in Fig. 6c and the fitted ice surface, the improved CCR*

*elevation (black square) was calculated. After that, the photons were averaged within each of pulse (pulse id from 19 to 31, blue dots in Fig. 6d) to fit another Gaussian function with an $R^2$ of 0.9214 and an RMSE of 1.6 cm. The peak position of the Gaussian function was used as the representative photon of the CCR that has an offset of 107.0 cm in the horizontal direction and 36.0 cm in the vertical direction from the estimated CCR location."*

**l.333 As discussed on l.264, I don't believe that the difference in the offset between ALT03 and ATL06 is significant.**

Now we filled the data gaps of GT2L and GT2R in Table 2. The difference between the bias of 1.5 cm for ATL06 and 4.3 cm for ALT03 still exists. However, considering their standard deviations of 9.1 cm and 8.5 cm, we agree with you that this difference is not significant.

In the **Results** section we revised it to: "…… *The difference between the bias of 1.5 cm for the processed ATL06 application product (L3A Land Ice Height data) and that of 4.3 cm for the unprocessed ATL03 product (L2A Global Geolocated Photon Data) is considered insignificant, taking their precision values, 9.1 cm and 8.5 cm, respectively into account.*"

---

## Author Comment (AC3) · 23 Mar 2021

We thank all three reviewers for their constructive comments and suggestions. This manuscript will be much improved by their input. We have made changes to our manuscript. In the following responses, we use "**bold**" text for reviewer's comments, "non-bold" text for our responses, and "*italic*" for changed text in the manuscript.

**Referee #3**

**1 Overview**

**Li et al. (2020) deliver results from an Antarctic campaign designed to assess elevation measurements from NASA's Ice Cloud and land Elevation Satellite-2 (ICESat-2). For the most part, this was a well-designed and well-thought-out experiment for evaluating the ICESat-2 data. The work presented by the authors falls within the scope of The Cryosphere and could make a good contribution for ICESat-2 calibration and validation (cal/val). Overall, while this is a well-written manuscript, there are a few issues that should be resolved before its publication.**

Thanks. The following is our responses. Changes are made accordingly in the manuscript.

**2 Broad comments**

- **The cal/val data from the CHINARE campaign needs to be publicly accessible to be of use in the ICESat-2 project science office and the scientific community**

The data set is now uploaded to a publicly accessible site Dryad at https://datadryad.org/stash/share/mU52z7OSWAG07tTsGCWXt0Rm0MmT12qwdPORBIUpsnw. They include a) traverse validation data (GNSS data of traverse crossovers, GNSS – ICESat-2 crossovers and ice surface heights), b) CCR and RTS validation data (GNSS CCR positions and GNSS RTS check points), and c) DEM data (DEM and check points).

- **There are other supporting and cal/val efforts that should be mentioned in the text (e.g. NASA Operation IceBridge, Greenland Summit Station (Brunt et al., 2017), salar de Uyuni (Borsa et al., 2019), and the updated Antarctic pole hole campaign (Brunt et al., 2021)**

The suggested work and references are now added to **Introduction**: "*……A comprehensive validation of the surface elevations of the previous ICESat mission using ground GNSS observations was performed on the salar de Uyuni in Bolivia, which also estimated the inter-campaign biases occurred between different campaigns during the mission (Borsa et al., 2019). Prior to ICESat-2 launch, calibration and validation experiments were conducted on both the Greenland and Antarctic ice sheets (Brunt et al., 2017 and 2019b; Magruder and Brunt, 2018). The annual Antarctic campaigns traverse a 300 km stretch of the interior of Antarctica near 88ºS covering 20% of the ICESat-2 reference ground tracks (RGTs) (Brunt et al., 2019b and 2021)……The achieved geolocation accuracy ranges from 2.5 m for laser spot 6 to 4.4 m for laser spot 2 (Luthcke et al., 2021)……*".

- **The L/R designations of the ICESat-2 beams do not correspond with weak/strong full time as it depends on the orientation of the spacecraft. Might also help to include statistics on laser spots (1–6) to help determine any drift or biases in a given beam.**

We discussed this issue and added the correspondence between "left and right" and "laser spot" IDs in the Method section: "……*The left and right correspondence in Fig. 5a may change as the spacecraft changes its orientation. They are also referred to as reference ground tracks (RGTs) with laser spot IDs (1, 2…, 6) corresponding to six beams which are independent of spacecraft orientation. During our study period, the correspondence is (Laser spot 1: 3R), (Laser spot: 3L), (Laser spot 3: 2R), (Laser spot 4: 2L), (Laser spot 5: 1R), and (Laser spot 6: 1L) (Neumann et al., 2019)......*"

The laser spot IDs are also added into Table 2 (also Table B1).

| Ground track (Laser spot ID) | ATL06 Bias ± Precision (cm) | ATL03 Bias ± Precision (cm) |
| --- | --- | --- |
| GT1L (Laser spot 6) | +2.7 ± 9.6 (N = 64) | +5.9 ± 5.9 (N = 1518) |
| GT1R (Laser spot 5) | +3.0 ± 7.3 (N = 62) | +1.7 ± 6.7 (N = 2608) |
| GT2L (Laser spot 4) | +0.7 ± 7.9 (N = 48) | -0.5 ± 6.7 (N = 862) |
| GT2R (Laser spot 3) | - 2.3 ± 12.0 (N = 42) | +5.8 ± 14.0 (N = 1356) |
| GT3L (Laser spot 2) | +1.3 ± 8.4 (N = 33) | +4.2 ± 7.7 (N = 800) |
| GT3R (Laser spot 1) | - 0.7 ± 8.7 (N = 36) | +4.6 ± 10.9 (N = 2695) |
| ALL | +1.5 ± 9.1 (N = 285) | +4.3 ± 8.5 (N = 9839) |

We do not see any apparent drift or biases in a given beam.

- **While ICESat-2 will presumably help improve mass balance and sea level determination efforts with satellite altimetry, these are not in the mission requirements as they require modeling efforts**

We considered your comment here (as well as other two reviewers' comments) and lowered the significance of the results in **Introduction**: "…… *The validation results demonstrate that the estimated ICESat-2 elevations are accurate to 1.5–2.5 cm in this East Antarctic region, which shows the potential of the data products for eliminating mission biases by overcoming the uncertainties in the estimation of mass balance in East Antarctica......*"

- **Links included in the text should have labels for when the website was last date accessed. Some of these links can also be simplified by removing optional URL parameters.**

We added access dates and removing optional URL parameters.

**3 Line-by-line comments**

**Page 1, Lines 9–10: should probably be something like "We present the results of an assessment of ice surface elevation measurements from NASA's Ice Cloud and land Elevation Satellite-2 (ICESat-2) along the CHINARE (CHINese Antarctic Research Expedition) route near the Amery Ice Shelf**

**in East Antarctica.”**

We have replaced the sentence with the suggested one “*We present the results of an assessment of ice surface elevation measurements from NASA's Ice Cloud and land Elevation Satellite-2 (ICESat-2) along the CHINARE (CHINese Antarctic Research Expedition) route near the Amery Ice Shelf in East Antarctica……*”

**Page 1, Line 13: “. . . the ICESat-2 geolocated photon product (ATL03) and land ice elevation product (ATL06). . . ”**

We have changed the sentence accordingly.

**Page 1, Line 17 replace “in a previous study” with “(Brunt et al., 2021)”**

We have replaced the “*in a previous study*” with “*(Brunt et al., 2021)*”. It is also updated in **References**.

**Page 1, Lines 21–22 While this is an important study, it is limited to a small region of East Antarctica covering a small percentage of ICESat-2 reference ground tracks (RGTs). Need to be careful not to overstate the results here. Not sure how these results help overcome the uncertainties in East Antarctic mass balance.**

We considered yours and other two reviewers' comments here to lower the significance of the results: “…… *The validation results demonstrate that the estimated ICESat-2 elevations are accurate to 1.5–2.5 cm in this East Antarctic region, which shows the potential of the data products for eliminating mission biases by overcoming the uncertainties in the estimation of mass balance in East Antarctica……*”

**Page 1, Line 23 What do you mean by “especially during the later operation period”? Do you mean that such field capabilities cannot be implemented for a couple of years, or that it is important to calibrate against potential degradation of the satellite measurements?**

It is the latter case. We revised the text: “……*especially for calibration against potential degradation of the elevation measurements during the later operation period*.”

**Page 1, Lines 25–27 This sentence is awkwardly phrased.**

It is rephrased: “*The new photon-counting laser altimetry satellite, Ice, Cloud, and Land Elevation Satellite-2 (ICESat-2), was successfully launched by the National Aeronautics and Space Administration (NASA) on September 15, 2018 (National Research Council, 2007; Markus et al., 2017; Neumann et al., 2019)……*”

**Page 2, Line 30 ATLAS is the primary instrument along with the GPS transceivers and the star cameras.**

We replaced “*single*” with “*primary*”: “…… *The primary instrument onboard ICESat-2, the Advanced Topographic Laser Altimeter System (ATLAS), is a photon-counting laser altimeter using 532 nm wavelength laser pulses ……*”

**Page 2, Lines 33-34 ICESat-2 will likely help improve mass balance and sea level contribution estimates from satellite altimetry, but those are not part of the mission requirements.**

The sentence is revised to delete that specific part: “…… *which is designed to conduct surface-elevation*

*observations at centimeter-level accuracy*  *(Markus et al., 2017; Neumann et al., 2018, 2019).*"

**Page 2, Line 34 The 0.4 cm/yr target is a mission requirement, not the current state of knowledge or uncertainty in elevation change.**

Thanks for the comment. We decided to delete this sentence and leave this for future papers to prove its realization: "."

**Page 2, Lines 34–37 This sentence is awkwardly phrased. Could be something like "We use Release-3 of the ICESat-2 geolocated photon elevation (ATL03) and land ice surface elevation (ATL06) products provided by the US National Snow and Ice Data Center (NSIDC)."**

It is revised: "……*We use Release 003 of the ICESat-2 geolocated photon elevation (ATL03) and land ice surface elevation (ATL06) products provided by the US National Snow and Ice Data Center (NSIDC) (NSIDC, 2021; Neumann et al., 2019; Smith et al., 2019)*".

**Page 2, Lines 38–39 "The calibration and validation of measurements is important for all satellite missions, particularly for missions with new instruments or technology, such the photon-counting laser altimeter on-board ICESat-2."**

The sentence is revised accordingly: "*The calibration and validation of measurements are important for all satellite missions, particularly for missions with new instruments or technology, such as the photon-counting laser altimeter on-board ICESat-2……*"

**Page 2, Lines 42–43 "Before launch, the ICESat-2 Project Science Office (PSO) funded calibration and validation experiments to be conducted on both the Greenland and Antarctic ice sheets. The annual Antarctic campaigns traverse a 300km stretch of the interior of Antarctica near 88° S covering 20% of the ICESat-2 reference ground tracks (RGTs) (Brunt et al., 2019)."**

They are changed accordingly: "*…… Prior to ICESat-2 launch, calibration and validation experiments were conducted on both the Greenland and Antarctic ice sheets (Brunt et al., 2017 and 2019b; Magruder and Brunt, 2018). The annual Antarctic campaigns traverse a 300 km stretch of the interior of Antarctica near 88ºS covering 20% of the ICESat-2 reference ground tracks (RGTs) (Brunt et al., 2019b)……*"

**Page 2, Line 45 "the Antarctic Ice Sheet (AIS)"**

It is changed accordingly.

**Page 2, Lines 47–48 "The NASA-led team also placed and used corner cube retroreflectors (CCRs) to collect ICESat-2 signatures at known points to help determine the horizontal geolocation accuracy of the laser pointing determination."**

We changed it to: "……*The NASA-led team also placed and used corner cube retroreflectors (CCRs) to collect ICESat-2 signatures at known points and determined the horizontal geolocation accuracy of the laser pointing*

*as 2 - 5 m (Magruder et al., 2020), specifically ranging from 2.5 m for beam 6 to 4.4 m for beam 2 (Luthcke et al., 2021)."*

**Page 2, Line 51 Again, while this is an important study, this is not a complete study of the "whole" of Antarctica.**

We changed it to "…… *additional coverage containing the lower-latitude interior and coastal regions in AIS should make the validation of ICESat-2 data complete with an ample and comprehensive understanding of elevation of diverse regions of*  *AIS.*"

**Page 2, Line 56 Replace "mass" with "volume"**

Considering that "mass balance" has a context meaning here, we would like to keep "mass", but add "volume" in parentheses: "…… *mass (volume) balance of AIS…...*"

**Page 2, Line 70 horizontal or vertical accuracy?**

We changed it to: "…… *with centimeter accuracy (vertical)*".

**Figure 1 Where is the inset map of Great Wall Station located? Does the inset need to be included if the station was not used as part of the campaign?**

We deleted "Great Wall Station" in the inset.

[Figure]

**Figure 1. ICESat-2 validation campaign based on coordinated multi-sensor ground observations along the 36th CHINARE route. The Landsat image mosaic of Antarctica (Bindschadler et al., 2008) is used as background.**

**Table 1 This is less ICESat-2 data than I would have thought. Are these numbers reduced using quality flags?**

(Table 1) The numbers of ICESat-2 observations listed in the table depend on the way we select ICESat-2 points (photons) for comparison with the GNSS points. As explained in the **Method** section, each ATL06 point is compared with at least 5 GNSS points within a neighborhood of 20 m, plus the "*satl06_quality_summary*" tag must be 0 (best quality). The total number of ATL 06 points listed in Table 1 are those paired points after reduction using the quality tag. In addition, the ATL 03 photons listed in Table 1 are those within a neighborhood of 20 m of the ICESat-2 – GNSS crossovers after a reduction using "*signal_conf_ph*" = 1, 2, 3 and 4 (buffer - high).

**Table 1 What do you mean by "not applicable" for ATL06 geolocation accuracy? That these are simply parameterized in the product?**

(Table 1) The elevations of ATL 06 points are representative elevations of 40 m segments and assigned to the segment centers. Therefore, their horizontal location accuracies may not be as meaningful as those for the ATL03 photons. So, we now leave the geolocation accuracy for ATL 06 blank in Table 1.

**Page 5, Line 97 Why were 6 cm CCRs chosen for this campaign?**

We add the text in the **Data** section to explain why we used 6 cm diameter CCRs: "*Under the time constraints of equipment shipment before expedition and manufacturing cycle of new products we used ten readily available CCRs of 6 cm diameter (Fig. A1a) at each site, which were designed for ground - based laser distance measurement. They were placed linearly at a 10 m interval across a nominal ICESat-2 ground track to reflect photons from ATLAS (Fig. 2a)……*"

We added a paragraph in the **Discussions** section to explain the impact:

"*The use of the readily available CCRs of 6 cm diameter for the 532 nm wave length of ATLAS, which is larger than 8 mm of the CCRs used in Magruder et al. (2020), is subject to velocity aberration caused by a decreased central disc and receiving signals from the outer lobes of the Fraunhofer diffraction pattern (Chang et al., 1971; Born et al., 1999; Sun et al., 2019; Magruder et al., 2020). In addition, the larger aperture of the CCR resulted in a higher level of the total signals received by ATLAS so that signals from both the smaller central disc and outer lobes are detected and used to estimate elevations in ATL03 data. This may have attributed to the creation of the long along-track streaks of ~35 m (Fig. 6a) and ~38 m (Fig. 6b) in comparison to those of ~11 m in Magruder et al. (2020). Thus, photons reflected from the lower neighboring CCR(s) in the cross-track direction (Fig. 6b) were detected for the same reason because of the symmetric Fraunhofer diffraction pattern. Similarly, the one-layer photon streak (green dots in Fig. 6a) may include those reflected from one or both neighboring CCRs because the elevations of all three neighboring CCRs (#4, #5 and #6) are within a 15 cm range (Table C1) due to local ice surface topography and logistic constraints, although the poles were manufactured in different lengths. On the other hand, the received signals in the central disc are generally of a higher level (about 84% of the total energy) than in the outer lobes given atmospheric scattering and other optical losses (Magruder et al., 2020). Correspondingly, we observe that within the window of the nadir CCR (red rectangle in Fig. 6b) the photons are densely aligned along a curve. The same curve trend appears to continue towards both ends, but diverged by potentially blended signals reflected from neighboring CCRs (Figs. 6a and 6b). Therefore, by selecting photons inside the central window of the CCR streak it ensures that*

*high quality photons in the central disc of the Fraunhofer diffraction pattern be used to estimate the elevation of the representative photon of the nadir CCR through the fitting curve. The result is also validated by the nadir CCR position surveyed by using the high-precision GNSS RTK technique.*"

[Figure]

**Figure 6 (revised version)**

We revised and added the text in the **Results** and **Method** sections to explain how we select the appropriate photons to analyze the data:

In the **Results** section "*……Among the CCRs, CCR #5 near Zhongshan Station and CCR #6 near Taishan Station were closest (~0.3 m and ~1.3 m) to the ground tracks and are called nadir CCRs. Given these CCR shifts from ground tracks, uncertainty of the ground tracks themselves of up to ~6.5 m (Magruder et al., 2020), and the CCR interval of ~10 m, we expected to have one to two CCRs falling into a footprint of ~11 m. Fig. 6b clearly shows returned photons (green dots) of two layers with an elevation difference of ~50 cm. Furthermore, the photons in the upper layer within a window of ~9 m (red rectangle in Fig. 6b) around the nadir CCR (#6) were received by ATLAS, while those of the lower elevation were not present inside the window. Therefore, we only use the returned photons inside a window of ~9 m around the nadir CCR to validate ICESat-2 elevations*".

In the **Method** section "*……where the photons are generally of high quality and have the confidence tag "signal_conf_ph" equal to 3 or 4 to calculate an average elevation in each pulse to reduce noises and potential atmospheric effect (blue dots in inset of Fig. 6a). The average elevations are then further fitted to a Gaussian curve. The peak of the curve is treated as the representative CCR photon position. An offset is then calculated between the positions estimated from the ATL03 photons and the GNSS survey.*"

We also answered a similar question asked by Reviewer 1. Please see details in the responses to Reviewer 1's comments with more figures and discussions.

**Page 5, Line 102 Possible to use a different positioning technique for these to reduce the impact of the station problems? 1 meter vertical is not going to be beneficial from a cal/val standpoint.**

Due to logistic difficulties, GNSS-measurements near Taishan Station were carried out using the single-point positioning technique. The GNSS function is integrated into a field surveying pad system. The data are not appropriate to support a PPP solution. We will look into a potential PPP application for these single point positioning situations in the future.

**Page 5, Line 109 "that are separated by 90 meters"**

"…… *that is ~90 m apart.*" is changed to "…… *that are separated by 90 meters.*"

**Page 5, Line 115 "We selected a silver-gray coating with R = 0.235 as it was the closest to the reportedly highest estimated probability (EP) of photon detection coating with R = 0.28"**

We accepted the suggestion and used the sentence in the text.

**Page 5, Line 109 Remove "Thus"**

It is removed.

**Page 8, Line 167 "which are 3 km apart"**

We have corrected the sentence: "…… *which are 3 km apart.*"

**Page 8, Line 168 Weak/Strong beams can be either left or right depending on the spacecraft orientation.**

We discussed this issue and added the correspondence between "left and right" and "laser spot" IDs. The text is revised accordingly: "……*The left and right correspondence in Fig. 5a may change as the spacecraft changes its orientation. They are also referred as reference ground tracks (RGTs) with laser spot IDs (1, 2…, 6) corresponding to six beams which are independent of spacecraft orientation. During our study period, the correspondence is (Laser spot 1: 3R), (Laser spot: 3L), (Laser spot 3: 2R), (Laser spot 4: 2L), (Laser spot 5: 1R), and (Laser spot 6: 1L) (Neumann et al., 2019)……*"

**Page 8, Line 171 "We reduce the impact of non-signal and noisy measurements by reducing the ATL06 land ice elevation measurements using the atl06_quality_summary flag and the ATL03 geolocated photon measurements to medium to high confidence photons using the signal_conf_ph flag. We also consider . . . "**

We accepted the suggestion and used the sentence in the text.

**Page 8, Lines 174–175 What determination did you use to decide if buffer to low classified photons should be included? In some cases, the buffer to low classified photons often can improve comparisons with ground measurements due to the shape of the ATLAS transmit pulse (which can be truncated in the ATL03 classifier if only including high-quality PEs).**

The sentence is revised: "……*We also consider the ATL03 photons with "signal_conf_ph" equal to 1 (buffer) and 2 (low) to reduce the effect of the transmit pulse shape bias that may be caused by truncation of the return pulse through exclusion of these lower confidence photons (Smith et al., 2019; Brunt et al., 2019b).*"

**Page 8, Line 177 Remove "On the other hand"**

The phrase is removed.

**Page 9, Lines 187-189 As this campaign includes multiple different terrains, was slope considered when comparing with the ATL06 measurements? i.e. along-track slope is estimated when calculating the**

**average surface height in ATL06. Would your ground measurements also provide a metric for the along-track and acrosstrack slopes estimated by ATL06?**

Thanks for bringing this up. Our data can only provide along - GNSS track slopes. We computed the GNSS slopes at each GNSS – ICESat-2 (ATL06) intersections using GNSS points within a 20 m radius window, so the slope scale matches that of the 40 m ATL06 segments. The GNSS slopes are the result of a least-squares linear fit to the GNSS points. In addition, we also calculated the intersection angle $\alpha$ (Fig. AC3-1)

[Figure]

**Fig. AC3-1**

We generated the following figures to compare the slopes between GNSS (along - GNSS track) and ATL06 (along - and cross – track) results. Fig. AC3-2a shows that the GNSS slopes are relatively consistent with the ATL06 along - track slopes ($R^2 = 0.49$); the RMSE between the two sets of slopes is 0.0062 (0.36°). In contrast, the comparison with the ATL06 across - track slopes resulted in a low $R^2$ of -0.03 and RMSE of 0.0089 (0.51°) (Fig. AC3-2c). Their histograms are presented in Figs. AC3-2b and AC3-2d. Note that we removed four ATL06 slopes because their error flags dh_fit_dy are too large (in the order of 3.4e+38). Considering that the intersection angles (α) are mostly within 30°, the ATL06 tracks are roughly aligned with the GNSS tracks. Therefore, the GNSS slopes are also closer to the ATL06 along – track slopes.

[Figure]

**Fig. AC3-2**

We performed a preliminary analysis of differences between the GNSS and ICESat-2 elevations vs. the intersection angle α at the crossovers. We did find some large elevation differences associated with increased α. Furthermore, a significant correlation between the elevation differences and along - track or cross – track slopes is not present (very preliminary).

**Page 9, Lines 194–196 The early mission pointing issue is a known problem (was due to a reference frame mismatch in the onboard software for the star cameras) (Luthcke et al., 2021).**

The sentence is revised: "*Our preliminary analysis of Releases 001 and 002 data indicated that the offsets between the actual and reference ground tracks in the study area were reduced from up to ~3000 m during the initial mission, which is caused by a reference frame mismatch in the onboard software for the star cameras (Luthcke et al., 2021), to 1–6 m before our expedition.*" The paper is also added to **References**.

**Page 10, Lines 206–212 These sentences are awkwardly phrased.**

The paragraph is now rewritten: "*The photons reflected from a CCR and received by ATLAS are represented as a streak of elevations along a track in the ICESat-2 ATL03 data (green dots in Fig. 6a above); they are distributed on both sides of the GNSS-surveyed location (black square). We select the photons in the central section of the streak, approximately one footprint long (within the red rectangle), as presented in the inset of Fig. 6a to estimate the representative CCR photon position. Then in each pulse the photons are selected using the confidence flag "signal_conf_ph" equal to 3 (good) or 4 (high) and averaged to reduce noises and potential atmospheric effect. The average elevations (blue dots) are further used to fit a Gaussian curve. The peak of the curve is treated as the representative CCR photon position. An offset is then calculated between the CCR positions estimated from the ATL03 photons (peak point) and the GNSS survey (black square).*"

**Page 10, Line 207 11 m laser footprint?**

We have added "~*11 m*": "*We select the photons in the central section of the streak, approximately one footprint long (~11 m, within the red rectangle) ......*"

**Page 10, Line 209 "confidence flag "signal_conf_ph" equal to middle to high confidence in order to calculate an average elevation in each pulse with reduced noise". You're calculating averages over individual pulses? Are there enough return PEs to calculate at this along-track length with significance? This is fitting at approximately 0.7 meters along track correct?**

Yes, this gives an average number of photons per pulse, spaced every ~0.7 m. For the two weak beam tracks at Zhongshan and Taishan stations, the CCRs returned a high number of photons: ~4 photons (middle - high confidence) per pulse at CCR, in comparison to ~2 photons (low - high confidence) per pulse on ice surface in the region. Therefore, there is a sufficient number of returned PEs to support the fitted Gaussian curve. The specific counts given are in the **Results** section.

In the 3$^{rd}$ paragraph of Section 4.2: "*From the returned photons near Zhongshan Station (Fig. 6a) we selected 51 photons of 13 pulses, 3.9 photons per pulse, located in the central part of the weak beam streak.*"

In the 4$^{th}$ paragraph of Section 4.2: "*CCR #6 was found to have returned 52 photons in 13 pulses, 4 photons per pulse, from the weak beam track within the ~9 m window in Fig. 6b.*" (Taishan Station)

**Page 10, Line 210 replace "further fitted" with simply "fit".**

We rewrote the sentence without "further": "*The average elevations (blue dots in inset of Fig. 6a) are used to fit a Gaussian curve.*"

**Page 10, Line 210 Are you using "e.g." here because you use different curves besides a Gaussian? What other functionals do you consider? Did you mean to use "i.e." here?**

We now directly used "*Gaussian curve*" (see above revised sentence).

**Figure 6 Is there a way of combining the plots for the same region to not repeat information? Maybe at some middle level of zoom?**

We modified Figure 6 accordingly (see Fig. 6 above).

**Page 11, Line 223 1.0 cm horizontal?**

It is elevation accuracy. Now both errors of GNSS observations and the reference point are considered. The elevation accuracy is 1.3 cm: "...... *Based on the internal precision given by the GNSS system and accuracy of the known GNSS reference point, the elevation accuracy of three GCPs surveyed by the RTK positioning technique is 1.3 cm.*"

**Page 12, Line 250 Can you clarify what is 816 meters apart? As phrased it could be interpreted as the GNSS measurements are over 800 meters from your ICESat-2 measurements.**

The sentence is revised to make it more specific: "*The average distance between the h$_2$ measurements and*

*GNSS – ICESat-2 intersections is ~2366 m.*" This average distance is increased from ~816 m to ~2366 m because the maximum distance is now extended from 2 km to 5 km after a test suggested by Reviewer 2.

**Table 2 Would be beneficial to have these statistics for laser spots in addition to the oriented beams (to directly map to individual laser beams).**

The laser spot IDs are added into Table 2 above.

**Page 12, Line 263 Again need to be careful to clarify that L/R do not necessarily map to weak/strong as it depends on the spacecraft orientation.**

Since the data gaps for Laser spots 3 and 4 (GT2L and GT2R) are now filled by using more qualified GNSS – ICESat-2 crossovers (see Table 2 above), these sentences are deleted.

**Page 13, Line 273 Replace "orbit" with RGT**

We have replaced "*orbit*" with "*RGT*".

**Page 13, Line 276 137 medium to high-quality classified photon returns?**

The sentence is revised accordingly: "*The ATL03 data showed a streak of 137 medium to high-quality classified photon returns in 50 pulses......*"

**Page 13, Line 284 1–2 meters vertical is not going to be accurate enough for cal/val purposes**

We agree with you. Thus, we only used the result of the CCR elevation comparison between the centimeter level accuracy GNSS and ICESat-2 observations at Zhongshan Station.

**Page 13, Lines 286–289 This seems possibly circular to use ICESat-2 to evaluate ICESat-2. What are the uncertainties in ATL06 here? What about horizontal geolocation errors of ATL06 and the CCR impacting the heights? What are the slopes?**

To avoid the impact of the horizontal geolocation errors of ATL06 on the elevation, we used the ATL03 photons to fit the ice surface (see Fig. 6 above), as also suggested by Reviewer 2. Although this may not solve the possibly "circular" issue, it demonstrates the need for precision GNSS survey at CCRs as long as logistics and field conditions can support.

Based on the slope flags in ATL06 segments, slopes in this section (Fig. 6b) range from -0.79° to -0.35°, with a mean and 1σ of -0.54° ± 0.18°. Although not significant, the ice surface slope in Fig. 6b appears steep because of the "squeezed" horizontal scale.

**Page 14, Lines 309–310 The weak beam also returns to 1/4th the number of detectors (4 instead of 16) and has different thresholds for saturation in the ATL03 algorithm.**

We may not have got your question quite right. Here is our try. The strong beams recorded 9.0 and 7.0 photons per pulse in average at the two sites (see table below): not saturated and no impact of CCR (90 m away). However, the weak beam, for example at Taishan Station, returned 9.7 photons per pulse in average. We further examined all 7 pulses over the 5 m - wide RTS sheet. Their photon counts per pulse are: 10, 12, 10, 9, 8, 9,

10. Correspondingly we can see three layers of returned photons (within the blue rectangle in AC3-3) from a) nadir CCR (top), b) neighboring CCR (middle) and ice surface (bottom), respectively. Since the travel times of these photons from different layers are different, many photons hit the four weak beam detectors at different times and thus, "managed" to be recorded by ATLAS between different dead times of the detectors. We think that it is how the high count-numbers per pulse (9.7 average) came about: saturated, but accepted and recorded by ATLAS.

[Figure]

**Fig. AC3-3**

**Page 14, Line 310 This is unfortunate that it was so close to the CCRs.**

We will avoid this situation in our potential future field experiments.

**Table 3 Should these return photon counts be in count/meter?**

We changed it to photon count per pulse averaged over the *5 m* segment of RTS and ice surface (firn).

| RTS coating (ICESat-2 beam, site) | Pre-expedition reflectivity | Avg count per pulse (RTS) | Full saturation fraction (RTS) | Avg count per pulse (firn) | Full saturation fraction (firn) |
|---|---|---|---|---|---|
| **Yellow** (strong beam, Zhongshan) | 0.532 | 9.0 | 0 | 6.1 | 0 |
| **Yellow** (strong beam, Taishan) | 0.532 | 7.0 | 0 | 6.7 | 0 |
| **Silver–gray** (weak beam, Zhongshan) | 0.235 | 1.1 (CCR) | 0.321 (CCR) | 1.7 | 0 |
| **Dark green** (weak beam, Taishan) | 0.060 | 9.7 (CCR) | 0.429 (CCR) | 1.7 | 0.072 |

**Page 14, Line 321 Replace "orbit" with "RGT"**

We have replaced "*orbit*" with "*RGT*".

**Page 15, Lines 341–343 Improving the $h_2$ measurements seems like a good advance for future campaigns.**

Thanks. It is on our to-do list.

**Page 17, Line 399 The cal/val data needs to be included here in the Data availability section**

The data set is now uploaded to a publicly accessible site Dryad at https://datadryad.org/stash/share/mU52z7OSWAG07tTsGCWXt0Rm0MmT12qwdPORBIUpsnw. They include a) traverse validation data (GNSS data of traverse crossovers, GNSS – ICESat-2 crossovers and ice surface heights), b) CCR and RTS validation data (GNSS CCR positions and GNSS RTS check points), and c) DEM data (DEM and check points).

**Page 21, Line 491–492 I believe that within a 20 meter ATL03 segment, only one photon event is directly geolocated in an absolute sense. The other PEs are then geolocated with respect to that reference PE.**

Thanks for the comment. The text is revised accordingly: "*Within a 20-meter along - track segment, only one photon, referred to as reference photon, is geolocated in an absolute sense. Other photons are then geolocated with respect to that reference photon (Neumann et al., 2019; Luthcke et al., 2021).*"

**References**

The following 5 suggested references are all cited in appropriate places in the manuscript and added to the reference list.

A. A. Borsa, H. A. Fricker, and K. M. Brunt. A Terrestrial Validation of ICESat Elevation Measurements and Implications for Global Reanalyses. IEEE Transactions on Geoscience and Remote Sensing, pages 1–14, 2019. ISSN 0196-2892. doi: 10.1109/TGRS.2019.2909739.

K. M. Brunt, R. L. Hawley, E. R. Lutz, M. Studinger, J. G. Sonntag, M. A. Hofton, L. C. Andrews, and T. A. Neumann. Assessment of NASA airborne laser altimetry data using ground-based GPS data near Summit Station, Greenland. The Cryosphere, 11(2):681–692, 2017. doi:10.5194/tc-11-681-2017.

K. M. Brunt, T. A. Neumann, and B. E. Smith. Assessment of ICESat-2 Ice Sheet Surface Heights, Based on Comparisons Over the Interior of the Antarctic Ice Sheet. Geophysical Research Letters, 46(22):13072–13078, Nov. 2019. ISSN 0094-8276. doi: 10.1029/2019GL084886.

K. M. Brunt, B. E. Smith, T. C. Sutterley, N. T. Kurtz, and T. A. Neumann. Comparisons of Satellite and Airborne Altimetry With Ground-Based Data From the Interior of the Antarctic Ice Sheet. Geophysical Research Letters, 48(2):2020–90572, Jan. 2021. ISSN 0094-8276. doi: 10.1029/2020GL090572.

S. Luthcke, T. Thomas, T. Pennington, T. Rebold, J. Nicholas, D. Rowlands, A. Gardner, and S. Bae. ICESat-2 Pointing Calibration and Geolocation Performance. Earth and Space Science, Feb. 2021. ISSN 2333-5084. doi: 10.1029/2020EA001494.

---

## Referee Report (RR1)

The authors have answered all my questions sufficiently and after restructuring the manuscript now is very clear and easy to follow. I have only a few remaining minor comments:

l.166 „GNSS data … were handled": Please modify, e.g. „GNSS data ...were processed"

l.174 „These crossovers are the intersections of tracks by the snowcat during instrument installations, observations, and overnight breaks." To clarify, I suggest a little modification: "...are the intersections of tracks by the snowcat which occurred usually during..."

l.188 „Tide-free" elevations are a convention. Applying all tidal corrections do not necessarily mean that the elevations are tide-free. However, Neumann et al. 2019 (p.111) informs that the elevations are given as „tide-free".

l.253 Why did you choose 30 as a number of sufficient pairs? If the ICESat and GNSS profiles are perpendicular, there can be 2 GNSS points and ~11 photon locations within a radius of 4m around the exact crossover location, which gives only 22 pairs. In this case, a perpendicular crossover would never be sufficient.

l.338 Please add a short notice about the reason for the significantly lower accuracy at Taishan here.

Furthermore, I have two suggestions for future works but I do not expect them to be included in this manuscript:

1. Concerning crossovers in the kinematic profiles, those mentioned at l.174 are within very short time differences, which implies that temporally correlated errors are similar in both segments. Do you also have any crossovers between the inbound and the outbound profiles? If not, this could be considered when planning such traverses in the future. Furthermore, you could install GNSS equipment an 2 independent snowcats and analyze crossovers between them.

2. You did a quite interesting analysis in response to my suggestion to use overlapping sections to access the accuracy. However, there seems to be a misunderstanding. To make my suggestion more clear I will explain it with a small drawing:

[Figure]

[Figure]

The upper part shows how you processed a section of the profile with a specific base station. I suggest to include some more observations to each section as in the lower part. This means that there is some overlap between the profiles (the observations in the red box) of different base stations. Now you can use the resulting coordinates in this overlap to analyze the accuracy and precision by calculating the mean offset and standard deviation between these different coordinate solutions.

---

## Referee Report (RR2)

**Assessment of ICESat-2 ice surface elevations over the CHINARE route, East Antarctica, based on coordinated multi-sensor observations**
Li et al.
The Cryosphere
https://doi.org/10.5194/tc-2020-330
12 May 2021

General Reviewer Comments:

Thank you for providing the revised manuscript. I believe it to be much improved from the original submission. However, I do have some comments for minor revisions to this version.

I believe figure 6 and the explanation of the CCR signals should be improved for more clarity to the reader.

- For 6b: The two CCR streaks at multiple elevations are explained by the different in height between #6 and #7. It would be better to plot them as different colors or label which signal returns are attributed to which optical component.
- For 6b: It is still unclear why there is a bump in the middle of the streak and the authors comments about high signal levels at nadir are not the reason unless they intend to imply that higher signal-to-noise levels make for higher elevations. Is it the slant range variation between t-approaching and t-nadir that is the difference? That can be calculated and compared to the apparent elevation increase for #6.
- Can the authors confirm that the higher SNR at t-nadir is truly the reason why returns were rejected from #7 during that central 9 m along track section?
- Why isn't the same signal bump present in figure 6a?

---

## Author Response (AR2)

**Responses to Editor and Reviewers' comments (minor revision)**

We thank editor and two reviewers for the constructive comments and suggestions.

**Editor**

**Line 43: Capitalize 'Salar'**

Done.

**Line 63: change to 'capability for the estimation of volume changes...' since part of the changes of the firn are unrelated to mass changes**

Done.

**line 204-205: the description of the RGT here is confusing, unclear what 'they' refers to. Your description seems to imply that there's one RGT for each beam pair, but that is not the case. See https://doi.org/10.5067/ATLAS/ATL06.003 for a description of the RGTs: "The Reference Ground Track (RGT) refers to the imaginary track on Earth at which a specified unit vector within the observatory is pointed. Onboard software aims the laser beams so that the RGT is always between ground tracks 2L and 2R (i.e. coincident with Pair Track 2)."**

It is revised: "*...... Furthermore, the reference ground track (RGT) is defined as an imaginary track between the nadir ground track pair (2L and 2R). All six laser beams then have laser spot IDs (1, 2..., 6)......*"

**Line 324-325: could you add information here about which beam track passed over the CCRs, as you did for Zhongshan station?**

The information is added: "*...... On the other hand, ICESat-2 passed across the CCR line array near Taishan Station along the weak beam track of 2L of RGT 0424 at 12:37 UTC on January 23, 2020 (green dots in Fig. 6b)......*"

**line 407: 'location, environmental conditions, and time'**

Done.

**line 438 'attributed to': do you mean 'contributed to'?**

It is changed to "*contributed to*".

**tc-2020-330-referee-report-1**

**The authors have answered all my questions sufficiently and after restructuring the manuscript now is very clear and easy to follow. I have only a few remaining minor comments:**

**l.166 „GNSS data … were handled": Please modify, e.g. „GNSS data ...were processed"**

Done.

**l.174 „These crossovers are the intersections of tracks by the snowcat during instrument installations, observations, and overnight breaks." To clarify, I suggest a little modification: "...are the intersections of tracks by the snowcat which occurred usually during..."**

Accepted and done.

**l.188 „Tide-free" elevations are a convention. Applying all tidal corrections do not necessarily mean that the elevations are tide-free. However, Neumann et al. 2019 (p.111) informs that the elevations are given as „tide-free".**

The sentence is changed to: "*…… Thus, the reduced ice surface elevations are given as "tide-free" (Neumann et al., 2019) and the permanent crustal deformation is removed (Schröder et al., 2017; Brunt et al., 2021)*"

**l.253 Why did you choose 30 as a number of sufficient pairs? If the ICESat and GNSS profiles are perpendicular, there can be 2 GNSS points and ~11 photon locations within a radius of 4 m around the exact crossover location, which gives only 22 pairs. In this case, a perpendicular crossover would never be sufficient.**

The threshold of 30 is adopted from Brunt et al., 2019b. In our study there are ~11 pules (instead of photons) within a radius of 4 m of a crossover, which gives 22 to 44 pairs of ICESat-2 – GNSS comparisons. The minimum case of 22 pairs may correspond to a perpendicular intersection. However, the most intersection angles between the ICESat-2 and GNSS traverses are around ~30 degrees, with the maximum of ~52 degrees. In fact, there is no ICESat-2 – GNSS pairs eliminated because of this 30 - pair threshold.

**l.338 Please add a short notice about the reason for the significantly lower accuracy at Taishan here.**

The sentence is changed to: "*…… Due to logistic difficulties, CCR positions were surveyed using the*

*single-point positioning technique at ~1 m accuracy level in both the horizontal and vertical directions.*"

**Furthermore, I have two suggestions for future works but I do not expect them to be included in this manuscript:**

**1. Concerning crossovers in the kinematic profiles, those mentioned at l.174 are within very short time differences, which implies that temporally correlated errors are similar in both segments. Do you also have any crossovers between the inbound and the outbound profiles? If not, this could be considered when planning such traverses in the future. Furthermore, you could install GNSS equipment an 2 independent snowcats and analyze crossovers between them.**

Thanks for these are very good suggestions. We will take them into account when planning the future campaigns.

**2. You did a quite interesting analysis in response to my suggestion to use overlapping sections to access the accuracy. However, there seems to be a misunderstanding. To make my suggestion more clear I will explain it with a small drawing:**

[Figure]

**The upper part shows how you processed a section of the profile with a specific base station. I suggest to include some more observations to each section as in the lower part. This means that there is some overlap between the profiles (the observations in the red box) of different base stations. Now you can use the resulting coordinates in this overlap to analyze the accuracy and precision by calculating the mean offset and standard deviation between these different coordinate solutions.**

Thanks for the clarification and the figures. We will look into this in a future extended analysis.

**Reference**

Brunt, K. M., Neumann, T. A., and Smith, B. E.: Assessment of ICESat-2 ice sheet surface heights, based on comparisons over the interior of the Antarctic ice sheet, Geophys. Res. Lett., 46(22), 13072-13078, https://doi:10.1029/2019GL084886, 2019b.

**tc-2020-330-referee-report-2**

**General Reviewer Comments:**

**Thank you for providing the revised manuscript. I believe it to be much improved from the original submission. However, I do have some comments for minor revisions to this version.**

**I believe figure 6 and the explanation of the CCR signals should be improved for more clarity to the reader.**

- **For 6b: The two CCR streaks at multiple elevations are explained by the different in height between #6 and #7. It would be better to plot them as different colors or label which signal returns are attributed to which optical component.**

  In Fig. 6b we labelled CCR IDs (#6 and #7) individually to each of the two layers.

- **For 6b: It is still unclear why there is a bump in the middle of the streak and the authors comments about high signal levels at nadir are not the reason unless they intend to imply that higher signal-to-noise levels make for higher elevations. Is it the slant range variation between t-approaching and t-nadir that is the difference? That can be calculated and compared to the apparent elevation increase for #6.**

  Our calculation indicated that the elevation difference caused by the slant range variation between t-approaching (20 m from nadir) and t-nadir, without considering the complex factors such as atmospheric, scattering, and other effects, is 0.8 mm, which would not make the bump height of 22.5 cm. With the data available at this point we do not have a clear answer as why this bump exists in Fig. 6b. It may also exist in Fig. 6a, but blended by the photons from the two neighboring CCRs. One possible reason, in addition to a "specific" atmospheric condition at that site and time etc., is that the large aperture may have created it. But this bump is symmetric at nadir position and with the averaged photon elevations the estimated peak position agreed with the precision GNSS position in this study (near Zhongshan Station). We will design a future experiment with different CCR aperture sizes to investigate the possible relationship between the aperture size and the bump (and other parameters).

- **Can the authors confirm that the higher SNR at t-nadir is truly the reason why returns were rejected from #7 during that central 9 m along track section?**

  We have examined the SNR values given in the ICESat-2 ATL06 data. "*SNR (estimated signal-to-noise ratio for the segment)*" flag accounts for the number of higher quality photons used for elevation fitting in each 40 m segment (Smith et al., 2020;

*/gtx/land_ice_segments/fit_statistics/snr)*. In Fig. b the three segments that contain photons of the central 9 m "single layer" section of the nadir CCR position near Taishan Station have the SNR of 31.61, 31.32, and 25.39, in comparison to the average SNR of 16.89 of the nearby firn surface. The SNR differences are significant, although they cannot be used to directly confirm the rejection of the photons from the neighboring CCR (#7).

Given the orbital configuration, CCR size and ground setting, and CCR streak data, the Fraunhofer refraction pattern analysis we have provided in the last round of responses to reviewers' comments showed a theoretical conclusion of the absence of CCR #7 photons in the central 9 m section due to the high SNR. However, based on the above analysis a numerical confirmation would have to come from additional signals in the original data that may or may not be saved onboard and stored in the mission data beyond ATL03. We hope to have in-depth discussions with the mission team and plan to conduct a future CCR experiment to achieve a numerical confirmation.

[Figure]

- **Why isn't the same signal bump present in figure 6a?**

Please see response to the second comment.

**Reference**

Smith, B., Hancock, D., Harbeck, K., Roberts, L., Neumann, T., Brunt, K., Fricker, H., Gardner, A., Siegfried, M., Adusumilli, S., Csathó, B., Holschuh, N., Nilsson, J., and Paolo, F.: Algorithm

Theoretical Basis Document (ATBD) for Land Ice Along-Track Height Product (ATL06), Goddard Space Flight Center Greenbelt, Maryland, 2020.

---

## Author Response (AR3)

**Response to Editor's comment (minor revision)**

**On line 442, change to "Although we cannot conclusively say so, this increased "signal-to-noise" ratio may have..."**

Thanks for the suggestion. It is changed to: "*……Although we cannot conclusively say so, this increased "signal-to-noise" ratio may have caused presence of the nadir CCR elevations only in the ~9 m central section along the photon streak in Fig. 6b (red rectangle)……*"